# Forward models demonstrate that repetition suppression is best modelled by local neural scaling

Arjen Alink[1], Hunar Abdulrahman[2] & Richard N. Henson [ID] [2]

Inferring neural mechanisms from functional magnetic resonance imaging (fMRI) is challenging because the fMRI signal integrates over millions of neurons. One approach is to compare computational models that map neural activity to fMRI responses, to see which best predicts fMRI data. We use this approach to compare four possible neural mechanisms of fMRI adaptation to repeated stimuli (scaling, sharpening, repulsive shifting and attractive shifting), acting across three domains (global, local and remote). Six features of fMRI repetition effects are identified, both univariate and multivariate, from two independent fMRI experiments. After searching over parameter values, only the local scaling model can simultaneously fit all data features from both experiments. Thus fMRI stimulus repetition effects are best captured by down-scaling neuronal tuning curves in proportion to the difference between the stimulus and neuronal preference. These results emphasise the importance of formal modelling for bridging neuronal and fMRI levels of investigation.

[1] University Medical Centre Hamburg-Eppendorf, Department of Systems Neuroscience, Martinistr. 52, 20246 Hamburg, Germany. [2] Medical Research Council, Cognition and Brain Sciences Unit, University of Cambridge, 15 Chaucer Road, Cambridge CB2 7EF, UK. Correspondence and requests for materials should be addressed to A.A. (email: a.alink@uke.de)

nferring the properties of neurons from gross measurements like functional magnetic resonance imaging (fMRI) is notoriously difficult. Because the fMRI signal integrates over millions of neurons, such inferences represent an inverse problem with no unique solution. However, progress can be made by comparing different "forward" models that formalise the mapping from neural activity to fMRI responses. This problem applies for example when trying to understand the phenomenon of reduced fMRI responses associated with repeated stimulation—referred to as "fMR adaptation" or "repetition suppression"[1–11]. These fMRI reductions have been attributed to a range of different neural mechanisms, such as "sharpening", whereby repetition is thought to narrow neural tuning-curves[12,13]. Here we address this problem by implementing a range of forward models that map from neuronal firing to fMRI signals, and comparing their predictions to key empirical features of fMRI repetition effects.

A previous study by Weiner and colleagues[14] formally modelled the relationship between neuron-level and voxel-level repetition effects. This study was an important demonstration that neural scaling and neural sharpening can reproduce similar effects of repetition on the mean fMRI response across voxels (a "univariate" response). They also assessed effects of repetition on fMRI activation patterns ("multivariate" responses). However, they did not determine whether multivariate repetition effects were best modelled by neural scaling or neural sharpening. This question is especially relevant given the recent claim by Kok et al.[15] that the prediction of upcoming stimuli (which may also arise from repetition) causes neural sharpening based on the observation that prediction improves fMRI pattern-based stimulus classification. Here we overcome the limitation of these studies by considering a wider range of neural models and by evaluating a larger combination of both univariate and multivariate fMRI data features. This enables us to determine, for example, whether repetition effects on fMRI pattern information are specific to neural sharpening.

Our forward models are based on four mechanisms associated with neuronal adaptation, each of which is grounded in previous neurophysiological studies: (1) scaling, where adaptation reduces response amplitude[16–18], (2) sharpening, where adaptation tightens tuning-curves[10], (3) repulsive shifting, where the peak of tuning-curves moves away from the adapting stimulus[19] and (4) attractive shifting, where the peak moves towards the adapting stimulus[20,21]. Each of these basic mechanisms is applied across three domains: (1) global, where all tuning-curves in a voxel are affected, (2) local, where tuning-curves close to the adapting stimulus are affected most, and (3) remote, where tuning-curves close to the adapting stimulus are affected least. Global effects could arise, for example, from neuromodulatory changes that affect a whole brain region; local effects could arise from activity-dependent changes such as synaptic depression[22]; while remote effects could arise from strengthening of inhibitory interneurons that implement "winner-takes-all" dynamics[23]. Figure 1 illustrates the 12 models resulting from crossing these four mechanisms and three domains. Tuning-curves are modelled by either Gaussian or von Mises distributions, and each model has only 2–3 free parameters: (1) the width of tuning-curves ($\sigma$), (2) the amount of adaptation ($a$) and, for non-global models, the extent of the domain of adaptation ($b$).

We compared the fMRI repetition effects predicted by these models to real fMRI data from two different experiments. Each experiment provided data for two stimulus-classes across multiple voxels within a single region-of-interest (ROI) in the brain. In Experiment 1, the data were responses from voxels in a fusiform face-responsive region (FFR) to initial and repeated brief presentations of face and scrambled face stimuli; in Experiment 2,

the data were responses in early visual cortex (V1) to initial and repeated sustained presentations of gratings with one of two orthogonal orientations. For each experiment, we examined how repetition affected six data features: (1) Mean Amplitude Modulation (MAM), the traditional univariate measure of repetition suppression, averaged across all voxels in the ROI; (2) Within-class Correlation (WC), the mean correlation of multivariate patterns across voxels between all pairs of stimuli of the same class; (3) Between-class Correlation (BC), the mean pattern correlation between all pairs of stimuli from different classes; (4) Classification Performance (CP), here operationalized as the difference between WC and BC, but validated by a Support Vector Machine (see Methods); (5) Amplitude Modulation by Selectivity (AMS), where the repetition-related change in amplitude is binned according to the selectivity of each voxel (defined as the $T$-value when contrasting the two stimulus classes); and (6) Amplitude Modulation by Amplitude (AMA), where the repetition-related amplitude change was binned according to the overall amplitude of each voxel. Some of these data features differed across our two experiments, presumably owing to important differences in ROI, stimulus type, stimulus duration, etc. To foreshadow our results, while no data feature on its own is diagnostic of a specific neural model, only the local scaling model can simultaneously reproduce all six data features using the same parameter values, and this is the case for both experiments, despite their different paradigms.

## Results

**Empirical results**. The two paradigms are illustrated in Fig. 2. Experiment 1 measured the impulse response to brief presentation (<1 s) of unfamiliar faces and scrambled faces that repeated immediately on 50% of occasions, in a randomly-intermixed, event-related fMRI paradigm[24]. Experiment 2 measured the sustained response to 14 s blocks containing rapid presentations of oriented visual gratings, with the orientation alternating between 45 and 135 degrees across blocks[25].

The six data features are shown for each experiment in Fig. 3. As expected, both Experiment 1 and Experiment 2 showed significant repetition suppression (MAM): in FFR ($t(17) = -7.53$, $p < 0.001$) and V1 ($t(17) = -7.13$, $p < 0.001$), respectively. Stimulus repetition also reduced both within-class (WC) and between-class (BC) correlations between trials in both experiments (Exp1/FFR, WC, $t(17) = -8.61$, $p < 0.001$, and BC, $t(17) = -5.84$, $p < 0.001$; Exp2/V1, WC, $t(17) = -7.19$, $p < 0.001$, and BC, $t(17) = -7.76$, $P < 0.001$). However, while the difference between repetition effects on within- and between-class correlations (CP) increased for V1 in Experiment 2 ($t(17) = +4.15$, $P < 0.001$), it decreased for FFR in Experiment 1 ($t(17) = -3.84$, $P = 0.0012$). In other words, repetition improved the ability to classify stimuli according to their two classes in Experiment 2 (consistent with the improved classification for predictable stimuli reported in Kok et al.[15]), but impaired such classification in Experiment 1 (as confirmed by support-vector machines in both cases). Furthermore, while linear regression showed that repetition suppression increased with mean amplitude (AMA) in both experiments (Exp1/FFR, $t(17) = +9.26$, $P < 0.001$; Exp2/V1, $t(17) = +7.83$, $P < 0.001$), its dependence on voxel selectivity (AMS) differed across experiments: increasing with selectivity in Experiment 1 (FFR, $t(17) = +3.46$, $p = 0.003$), but decreasing with selectivity in Experiment 2 (V1, $t(17) = -2.31$ $p = 0.034$). Thus, given their different ROIs, stimulus-types and stimulation protocols, it is interesting to note that there are both commonalities and differences in the effects of repetition across the two experiments.

To further check the generalisability of these results, we examined other ROIs that responded selectively in each

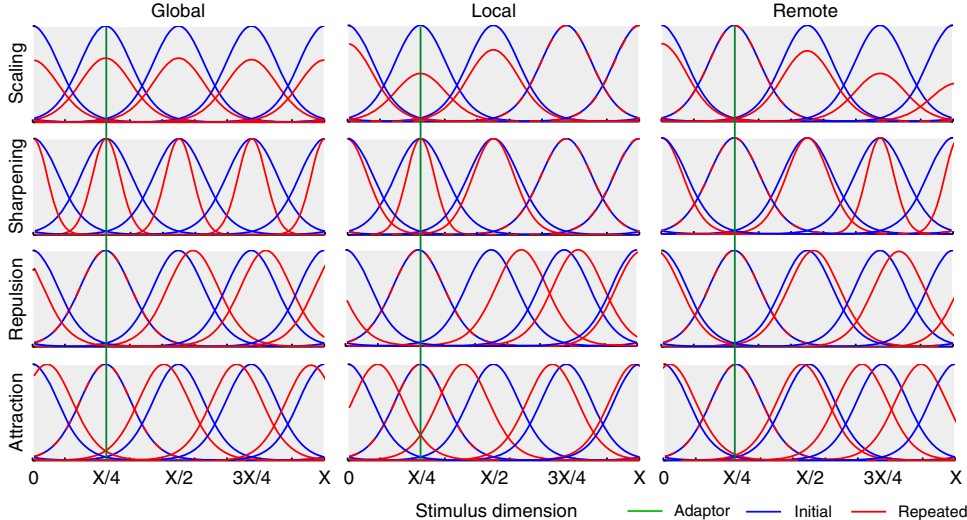

**Fig. 1** Example tuning curves along a stimulus dimension (ranging from 0 to X), both before (blue) and after (red) repetition of a single stimulus (with value X/4, shown by green line) according to the twelve different neural models of repetition suppression, created by crossing four mechanisms (rows) with three domains (columns). Mechanisms: scaling – adaptation reduces response amplitude, sharpening – adaptation tightens tuning-curves, repulsion – the peak of tuning-curves moves away from the adapting stimulus, attraction – the peak moves towards the adapting stimulus. Domains: global – all tuning-curves in a voxel are affected, local – tuning-curves close to the adapting stimulus are affected most, remote – tuning-curves close to the adapting stimulus are affected least. For illustrative purposes, only five neural populations are shown, equally-spaced along the stimulus dimension. Note that this figure illustrates the Gaussian tuning-curves used for Experiment 1 (see Supplementary Figure 1 for illustration of von Mises tuning-curves used for Experiment 2)

experiment. In Experiment 1, an occipital face-responsive region (OFR) showed the same pattern of data features as the FFR (Supplementary Figure 6), while in Experiment 2, both V2 and V3 showed the same pattern of data features as V1 (with the possible exception of AMS in V2, which was in the same direction but only borderline in significance, $p = 0.066$; Supplementary Figure 7). We also examined their generalisability across repetition lag in Experiment 1: delayed repeats showed the same pattern of significant data features in FFR as immediate repeats (just weaker in size, Supplementary Figure 8). Finally, because FFR responds more to faces than scrambled faces on average (unlike V1 which responds equally on average to different grating orientations), we split MAM and AMA data features by faces and scrambled faces and found the same pattern (Supplementary Figure 9), demonstrating that the effects are significant for both stimulus types, but just weaker for scrambled faces. Note that the other four data features depend on the difference between face and scrambled face responses, and therefore cannot be reported separately.

**Simulation results – unconstrained parameters**. We explored the predictions of each of the twelve models for each of the six data features in a grid-search covering a wide range of values for the 2–3 free parameters (see Methods for specific ranges used). We ran 50 simulations for each model for all unique parameter combinations. For each such combination, we calculated the 99% confidence interval across the 50 simulations for the mean of each of the six data features, and tested whether this was above, below or overlapped zero.

In this initial analysis, different parameter values were allowed for each data feature, in order to see whether any data features were diagnostic of a neural model, and whether each model could, in principle, explain each data feature. Figure 4 summarises the results for each experiment. Each circle represents a specific model, data feature and experiment. If there existed at least one parameter combination in which a model's 99% confidence interval for a feature was above zero, then that circle included red.

If there existed at least one parameter combination in which the confidence interval was below zero, then that circle included blue. If at least one combination produced a confidence interval that straddled zero, then that circle included white. When different parameter combinations produced two or more of these patterns, the circle was given a mixture of the corresponding colours.

The first thing to note is that no single data feature was sufficient to identify the underlying neural model, illustrating the difficulty of inferring from fMRI data at the level of voxels to mechanisms at the level of neurons (i.e., no value in any row in Fig. 4 is unique to one of the twelve models). Note that this conclusion holds regardless of the empirical value of the data features observed in the present experiments (leftmost column). This conclusion is important because some of these features, such as the increase in classification performance (CP) after repetition, have been assumed to support sharpening models[15], yet Fig. 4 shows that several other non-sharpening models can produce an increase in CP. The same goes for the negative slope in AMS, which was also thought to support sharpening models[15].

The second thing to note is that some of the neural models cannot produce at least one of the data features observed in the present experiments (whatever their parameter settings within the large range explored here). This can be seen by comparing the leftmost column with the remaining twelve columns. This means that, by considering a range of consequences of repetition (both univariate and multivariate), one can at least rule out some neural models. Nonetheless, with unconstrained parameters, there were six models that could fit the data features in Experiment 1 (FFR) – local and remote scaling, all three sharpening models and global repulsion – and three models that could fit the data features in Experiment 2 (V1) – local scaling, local sharpening and remote attraction.

**Simulation results – constrained parameters**. Figure 4 shows results based on parameters whose values were varied for each data feature separately (e.g., the values $\sigma$, $a$ and $b$ that produce a decrease in MAM may not be the same values that produce a

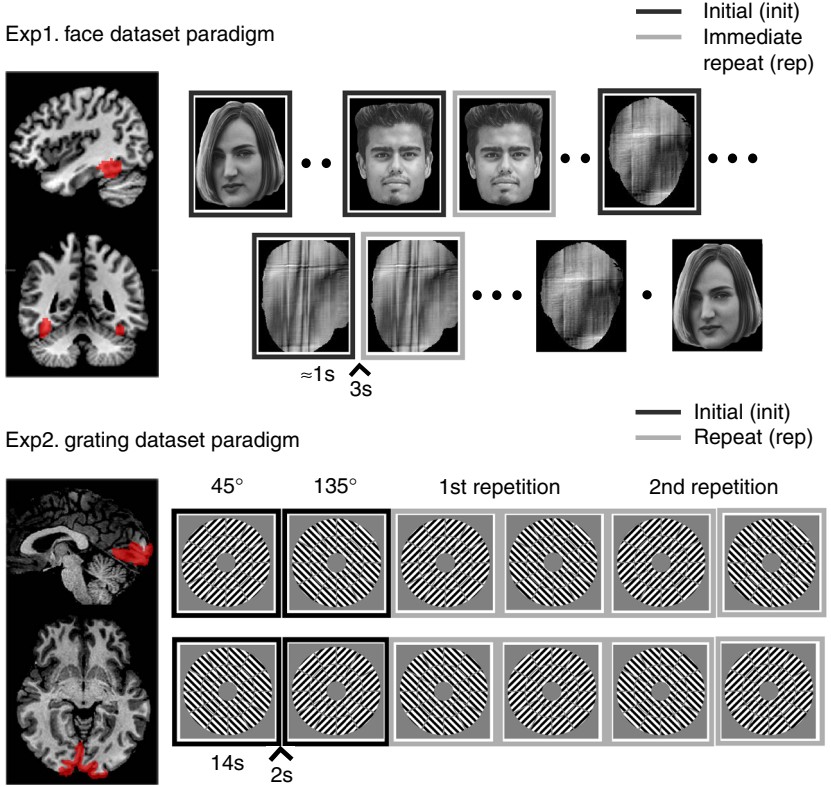

**Fig. 2** Visualisation of the two experimental paradigms and the corresponding brain region used in the analyses. Note that in our analyses we randomly dropped half of the initial trials in the Face dataset to balance the number of the initial and immediate repeats and that we used the 2nd repetition as the repeat condition for Experiment 2. The two face images shown are open license images presented for illustrative purposes only. The actual images used during Experiment 1 are described in Wakeman and Henson[24]

decrease in WC). When we constrained the parameters to have the same values across all six data features, only one model could simultaneously fit all six features: local scaling. This is shown in Fig. 5. If a circle is coloured green, then there existed a parameter combination in which that model's 99% confidence interval was consistent with the significant effect for that data feature (in that experiment); otherwise a circle is red. The particular parameter combination chosen was the one that simultaneously reproduced as many of the six data features as possible. Thus only if at least one parameter combination could simultaneously reproduce all six features would a whole column of Fig. 5 be green.

Only the column for the local scaling model has green colours for all data features. Moreover, this was the same model for both Experiment 1 and Experiment 2. The range of parameter values required for local scaling can be found in the Supplementary Table 1, and an illustrative fit to the data features can be found in Supplementary Figure 4 and the Supplementary Table 1. Therefore, the constraints offered by simultaneously fitting six different data features are sufficient to single out one of the twelve models considered here, and this was the same model across two independent experiments (just with different parameter values needed to capture differences between representations of the different stimuli across different ROIs, different stimulation protocols, etc.).

## Discussion

Several neural mechanisms have been hypothesised to underlie the observation of reduced fMRI responses to repeated stimulation, i.e. repetition suppression[3,12,13,26]. These include neural habituation or fatigue, which down-scale neuronal firing rates, and neural tuning or sharpening, which tighten neuronal tuning curves[3]. In principle, any of these mechanisms can explain the basic effect of repetition on the mean univariate fMRI response across voxels. However, we show that by (1) considering a range of features of fMRI repetition effects, both univariate and multivariate, and (2) formally modelling a range of potential neural mechanisms, various hypothetical neural mechanisms can be distinguished. Indeed, our results show that local scaling of neuronal firing is the only model, of the twelve considered here, that can simultaneously explain six features of repetition in fMRI data from two independent experiments. Importantly, local scaling at the neuronal level can explain sharpening of patterns at the voxel level (e.g. leading to improved classification after repetition), i.e., conceptually, sharpening at the voxel level does not imply sharpening at the neural level. This insight enables us to reconcile fMRI repetition effects with claims that tuning width of neurons in Macaque IT are unaffected by repetition[27], and emphasises the importance of formal modelling to bridge the differences between the neuronal and fMRI levels of investigation.

Our modelling also allowed us to demonstrate that no single feature of fMRI repetition effects was diagnostic of any of our 12 models (allowing for the 2–3 degrees of freedom corresponding to the parameters of each model). This reinforces the importance of simultaneously considering multiple data features. The local scaling model was the only model capable of simultaneously fitting our six data features with the same set of parameter values. This was true across datasets that differed in terms of the stimuli, brain region of interest, stimulation protocol and analysis method (randomly intermixed events vs. sustained blocks of repetition). It is possible that these findings could be explained by combinations of mechanisms (e.g. global scaling and local sharpening), or by

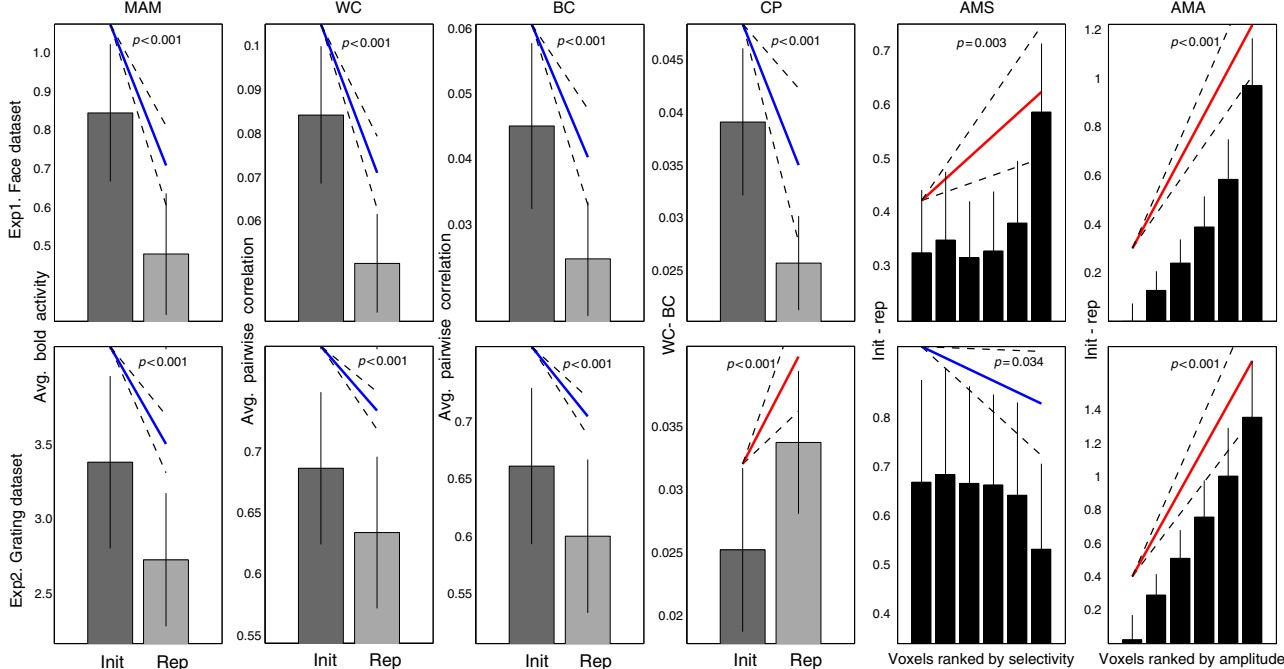

**Fig. 3** The six data features (columns) for each paradigm (rows). Data features: Mean Amplitude Modulation (MAM), Within-class Correlation (WC), Between-class Correlation (BC), Classification Performance (CP), Amplitude Modulation by Selectivity (AMS) and Amplitude Modulation by Amplitude (AMA). Bars reflect mean of each condition (init initial presentation, rep repeated presentation), with error bars reflecting 95% confidence interval given between-participant variability; diagonal line represents slope of linear contrast across conditions (red = positive; blue = negative) with dashed error margins reflecting 95% confidence interval of that slope (equivalent to pairwise difference when only two conditions)

neuronal mechanisms beyond the twelve considered here. Interestingly, our results show that fMRI experiments with just two levels for each stimulus type still enable one to differentiate between a wide range of possible models based on continuous neural tuning curves. However, it is possible that more than two stimulus levels will be needed to test simulations of repetition-related changes in neural tuning curves in future studies, e.g., to detect more subtle differences between different types of local scaling. Nonetheless, local scaling remains the most likely current explanation, in terms of parsimony.

A study by Weiner and colleagues[14] also used formal neural models to examine fMRI repetition suppression. They assessed the validity of two models—comparable to our global scaling and global sharpening models—and found that the validity of each model depended both on brain area and lag between stimulus repetitions. Neither global scaling nor global sharpening however was able to account for the full range of repetition effects in our two fMRI datasets. Nonetheless, Weiner et al.'s investigations raise the important possibility that the neural mechanisms underlying fMRI repetition effects may vary with other factors, e.g., in brain regions beyond those considered here.

To our knowledge, only two studies[14,15] have previously examined a combination of univariate and multivariate fMRI data features to elucidate neural mechanisms of fMRI response reductions. Specifically, Kok et al.[15] observed that stimulus predictability reduced mean fMRI responses (MAM) while increasing fMRI pattern information about stimulus class (CP). This observation, in conjunction with their finding that fMRI response reduction was reduced for voxels with higher stimulus selectivity (AMS), was interpreted as evidence for (global) neural sharpening. For the grating experiment (Experiment 2), we observed a similar pattern of empirical results. Although we did not explicitly manipulate prediction, the clear block structure of repeating stimuli would have produced strong predictions for upcoming stimuli (the same was not true for the face experiment, which may

be why it showed the opposite pattern for AMS). Importantly however, our simulations revealed that this conjunction of data features does not, in fact, uniquely identify neural sharpening. This is consistent with single-cell recording studies that claim that neural scaling provides a better description of repetition effects in macaque IT than neural sharpening[27]. Nonetheless, our results do not directly contradict the sharpening claim of Kok and colleagues[15], in which the paradigm differs from that employed in the two current experiments; rather the models considered here would need to be applied to Kok et al.'s data to see if local scaling was again the only model able to explain the full pattern of results, or whether local sharpening was better in this case.

How is the local scaling model flexible enough to produce opposite effects of repetition on CP and AMS across Experiment 1 and 2? The critical parameter turns out to be $\sigma$, the width of the neuronal tuning curves. As can be seen in the Supplementary Table 1, the optimal $\sigma$ values ranged between 0.2 and 0.4 for Experiment 1, but between 0.4 and 1 for Experiment 2. When $\sigma$ is low, a greater number of voxels have a selectivity for one stimulus (by chance), and so when these are suppressed, there is a decrease in WC after repetition (because highly tuned voxels are suppressed more), but relatively less decrease in BC. On the other hand, when $\sigma$ is large, there are fewer selective voxels and hence these are suppressed less, and there is less reduction in WC after repetition and a relatively larger reduction in BC. This trade-off between WC and BC allows CP to decrease for the face-scrambled distinction in Experiment 1 (when $\sigma$ is low) but increase for the orientation distinction in Experiment 2 (when $\sigma$ is high). For AMS, when $\sigma$ is low and a greater number of voxels are highly selective, the effect of local scaling is to lower their selectivity ranking. Because such selective voxels also show more suppression, there is a positive dependency between suppression and selectivity. Conversely, when $\sigma$ is large, local scaling tends to increase the selectivity ranking of less selective neurons, so there is a negative dependency between suppression and selectivity.

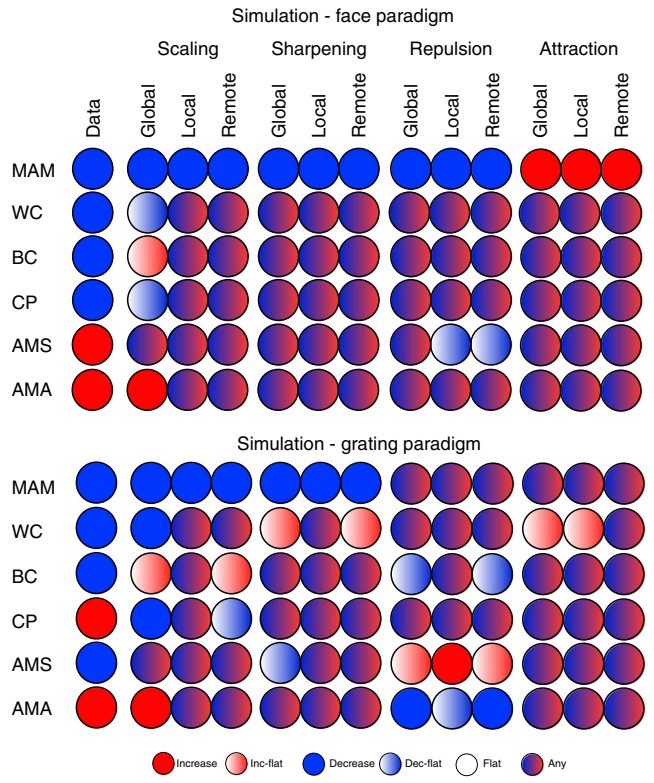

**Fig. 4** Possible simulated fMRI data features for all models (columns 2–13) for both experimental paradigms when considering all parameter combinations. The first column shows the empirical data features. Mechanisms: scaling – adaptation reduces response amplitude, sharpening – adaptation tightens tuning-curves, repulsion – the peak of tuning-curves moves away from the adapting stimulus, attraction – the peak moves towards the adapting stimulus. Domains: global – all tuning-curves in a voxel are affected, local – tuning-curves close to the adapting stimulus are affected most, remote – tuning-curves close to the adapting stimulus are affected least. Data features: Mean Amplitude Modulation (MAM), Within-class Correlation (WC), Between-class Correlation (BC), Classification Performance (CP), Amplitude Modulation by Selectivity (AMS) and Amplitude Modulation by Amplitude (AMA)

Note that $\sigma$ reflects the tuning curve width relative to the range of possible stimulus values (which was fixed as X here for both experiments), so the differences between experiments could be a property of the different ROIs and/or the different stimuli used (the differences in CP and AMS could also reflect other procedural differences between the experiments, but it is less obvious how such differences would affect $\sigma$). In any case, it is interesting that a single parameter within a simple model can produce such a range of different qualitative outcomes at the level of fMRI analysis, again questioning any "inverse" inference one might be tempted to make from fMRI.

Our finding that local scaling best explains fMRI repetition suppression does not question previous findings of stimulus repetition effects on single-cell recordings[16,17,19–21]. As alluded to above, it is possible that multiple mechanisms operate in parallel, but in different neural populations or cortical layers, and that the dominance of the local scaling model for fitting fMRI data is simply due the greatest proportion of neurons exhibiting local scaling. It is also important to keep in mind that our fMRI signals are dominated by changes in local field potentials[28], rather than the action potentials in large pyramidal cells that are normally measured in single-cell studies. Unfortunately, our fMRI data cannot speak to the temporal dynamics of repetition effects, such

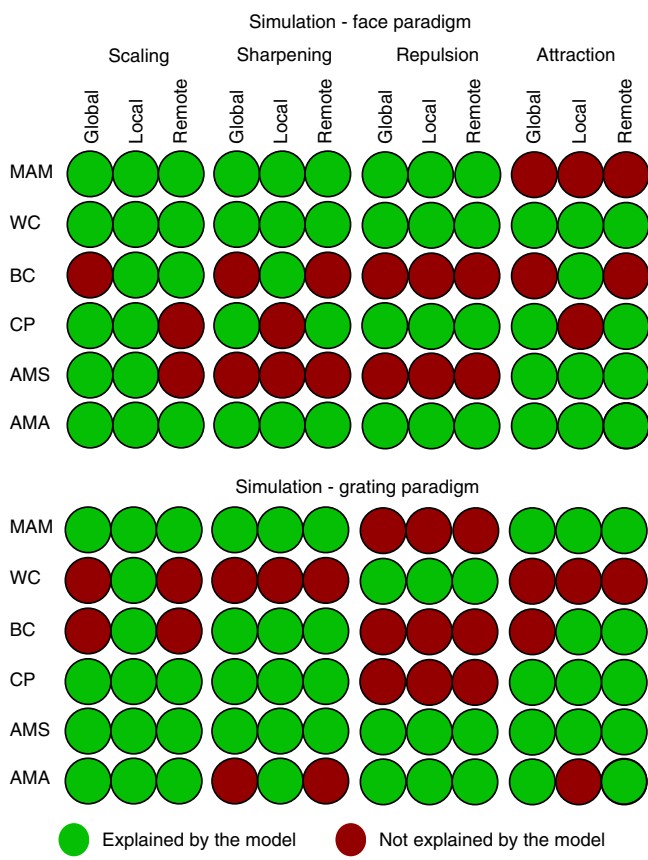

**Fig. 5** The maximum number of data properties explained by each model when parameters are constrained to be equal across all data properties (but can differ across experiments). Note that, for some models that can explain only 4 or 5 data properties with the same parameter values, there may be different subsets of the same number of data properties that can be explained (i.e., this figure only shows one such subset). Mechanisms: scaling – adaptation reduces response amplitude, sharpening – adaptation tightens tuning-curves, repulsion – the peak of tuning-curves moves away from the adapting stimulus, attraction – the peak moves towards the adapting stimulus. Domains: global – all tuning-curves in a voxel are affected, local – tuning-curves close to the adapting stimulus are affected most, remote – tuning-curves close to the adapting stimulus are affected least. Data features: Mean Amplitude Modulation (MAM), Within-class Correlation (WC), Between-class Correlation (BC), Classification Performance (CP), Amplitude Modulation by Selectivity (AMS) and Amplitude Modulation by Amplitude (AMA)

as the acceleration of evoked responses predicted by facilitation models[1] and neural synchronisation[11], which may also be important features of neural repetition effects.

Note that local scaling of neuronal tuning curves could itself arise from multiple potential mechanisms within the context of a neuronal circuit, such as synaptic depression of bottom-up inputs, or recurrent inhibition by top-down inputs. As reviewed by Vogels[18], firing-rate dependent response fatigue, e.g., a prolonged hyperpolarization that is intrinsic to the recorded neuron, is unlikely to explain the properties of the stimulus-specific repetition effects of the type described here. Indeed, when we simulated a simplified, two-parameter version of the local scaling model, in which the $b$ parameter was fully determined by the initial firing-rate of neurons, we could no longer reproduce the present repetition effects (see Fatigue model section of Methods). It is possible that the repetition effects observed here in FFR, OFR and even V1 are "inherited" from earlier stages in the visual

processing pathway (i.e, arise in the inputs to these areas[18]), which can potentially explain the need for a wider domain of adaptation (i.e, our additional $b$ parameter). Alternatively, the effects might arise from top-down input from regions later in the processing pathway. For example, the hypothesis of predictive coding, in which top-down predictions from higher-order regions are refined after repetition[13,29], could also result in maximal suppression of neurons that are most selective for the (repeated) stimulus—i.e., local scaling.

In sum, our work illustrates the value using forward models that map from neurons to voxels, like those considered here, to interpret fMRI data, i.e., map back from voxels to neurons. Despite the simplicity of the models considered here, their predictions are not always intuitive, and they therefore help protect against superficial analogies, for example that sharpening of multivoxel fMRI patterns entails the sharpening of neuronal tuning curves.

## Methods

**Ethics**. All participants gave their informed consent after being introduced to the experimental procedure, in accordance with the Declaration of Helsinki. Experimental procedures were approved by the Cambridge Psychology Research Ethics Committee (ethics reference numbers: CPREC 2005.08 and CPREC 2010.52 for Experiments 1 and 2 respectively).

**Experiment 1–participants and task**. We report data from 18 of the 19 participants (8 female, aged 23–37) described in Wakeman & Henson[24], after removing one participant who had fewer fMRI volumes than the others. In brief, participants made left-right symmetry judgments to faces or scrambled faces (no image was perfectly symmetrical, and the range of perceived degrees of symmetry was made apparent during a practice phase). Half of the faces were famous, but only the remaining unfamiliar faces were analysed here. Every stimulus was repeated either immediately or after a delay of 5–15 intervening stimuli; for our main analysis, only initial presentations and immediate repetitions were analysed, though we also analysed delayed repetitions in Supplementary Figure 8). For more details on the experimental paradigm see Wakeman & Henson[24].

**Experiment 1–fMRI acquisition and processing**. The MRI data were acquired with a 3 T Siemens Tim-Trio MRI scanner with 32-channel headcoil (Siemens, Erlangen, Germany). The functional data were acquired using an EPI sequence of 33, 3 mm-thick axial slices (TR 2000 ms, TE 30 ms, flip angle 78°). 210 volumes were acquired for each of the 9 sessions. Slices were acquired in an interleaved fashion, with odd then even numbered slices and a distance factor adjusted to ensure whole-brain coverage, resulting in a range of native voxel sizes of $3 \times 3 \times 3.75$ mm to $3 \times 3 \times 4.05$ mm across participants (for details see[24]). We also obtained a high-resolution (1 mm isotropic) T1-weighted anatomical image using a Siemens MPRAGE sequence for normalisation of brains.

The fMRI data were preprocessed using the SPM12 software package (www.fil.ion.ucl.ac.uk/spm) in Matlab 2012b (uk.mathworks.com). After removing the first two EPI images from each session to allow for T1 saturation effects, the functional data were corrected for the different slice times, realigned to correct for head motion, and coregistered with the structural image. The structural image was warped to a standard template image in MNI space, and the warps then applied to the functional data.

The main a priori ROI was the FFR, as defined by the group univariate contrast of unfamiliar faces vs. scrambled faces (averaged across initial and repeated presentations, and therefore not biasing analysis of subsequent repetition effects), thresholded at $p < 0.05$ family-wise error corrected using random field theory. (While this region is likely to overlap with the Fusiform Face Area (FFA) defined by Kanwisher and colleagues[30], the FFA is normally defined for individual participants using a wider range of non-face control stimuli). We extracted fMRI timeseries data from voxels combined across both left and right FFR. The combined masked contained 135 voxels for the right FFR and 50 voxels for the left FFR. The only other region that responded more to faces than scrambled faces at this threshold was a right Occipital Face-responsive Region (OFR). We also examined this region, but after lowering the threshold to $p < 0.001$ uncorrected to include left OFR as well (resulting in 75 and 193 voxels in left and right OFR respectively).

Since this design used a short SOA of between 2.9–3.3 s (randomly jittered), the BOLD response for each trial was estimated using the Least Squares Separate (LSS-N) approach[31,32], where $N$ is the number of conditions (qualitatively similar results were achieved with the standard Least Squares All approach). LSS-N fits a separate General Linear Model (GLM) for each trial, with one regressor for the trial of interest, and one regressor for all other trials of each condition (plus 6 regressors for the movement parameters from image realignment, to capture residual

movement artefacts). This implements a form of temporal smoothness regularisation on the parameter estimation[32]. The regressors were created by convolving a delta function at the onset of each stimulus with a canonical haemodynamic response function (HRF). The parameters for the regressor of interest for each voxel in the mask were then estimated using ordinary least squares, and the whole process repeated for each separate trial. The number of trials varied slightly from session to session but it was balanced across participants, totalling 49 trials for each of the 4 trial-types considered here (initial and immediate repetitions of unfamiliar and scrambled faces).

**Experiment 2- participants and task**. We report fMRI data from the visual gratings conditions of a larger study reported previously[22,29,30]. Eighteen healthy volunteers (13 female, age range 20–39) with normal or corrected-to-normal vision took part in the experiment.

The gratings were oriented 45° clockwise and 45° anticlockwise (135° clockwise) from the vertical, with a spatial frequency of 1.25 cycles per visual degree. These stimuli were presented during 2 runs of 8 min, with each run divided into 4 subruns, and each subrun containing 6 blocks, with the orientation presented in each block alternating (Fig. 1). Each block lasted 14 s and contained 28 phase-randomised gratings of one orientation, presented at a frequency of 2 Hz. The stimulus duration was 250 ms, followed by an interstimulus interval (ISI) of 250 ms, during which a central dot was present, surrounded by a ring that determined the task (see below). The spatial phase was drawn randomly from a uniform distribution between 0 and $2\pi$. Stimulus blocks were separated by 2 s fixation periods and subruns by 24 s fixation periods.

In addition, each participant participated in a 12-min run for retinotopic mapping. A description of the stimuli employed and the procedure used to define individual regions of interest (ROIs) for the primary visual cortex can be found in Alink et al.[25].

Participants were instructed to continuously fixate on a central dot (diameter: 0.06° visual angle). The dot was surrounded by a black ring (diameter: 0.20°, line width: 0.03°), which had a tiny gap (0.03°) either on the left or right side. The gap switched sides at random at an average rate of once per 3 s (with a minimum inter-switch time of 1 s). The participant's task was to continuously report the side of the gap by keeping the left button pressed with the right index finger whenever the gap was on the left side, and keeping the right button pressed with the right middle finger whenever the gap was on the right side. The task served to enforce fixation and to draw attention away from the stimuli.

**Experiment 2–fMRI acquisition and processing**. Functional and anatomical MRI data were acquired on the same scanner as Experiment 1 (see above). During each stimulus run, we acquired 252 volumes containing 31 transverse slices covering the occipital lobe as well as inferior parietal, inferior frontal, and superior temporal regions for each subject using an EPI sequence (TR = 2000ms, TE = 30 ms, flip angle = 77°, voxel size: 2.0 mm isotropic; field of view: 205 mm; interleaved acquisition, GRAPPA acceleration factor: 2). The same EPI sequence was employed for a retinotopic mapping run, during which we acquired 360 volumes. We also obtained a high-resolution (1 mm isotropic) T1-weighted anatomical image using a Siemens MPRAGE sequence.

Functional and anatomical MRI data were preprocessed using the Brainvoyager QX software package (Brain Innovation, v2.4). After discarding the first two EPI images for each run to allow for T1 saturation effects, the functional data were corrected for the different slice times and for head motion, detrended for linear drift, and temporally high-pass filtered to 2 cycles per run. The data were aligned with the anatomical image and transformed into Talairach space[33]. After automatic correction for spatial inhomogeneities of the anatomical image, we created an inflated cortex reconstruction for each subject. Our main a priori ROI was V1, given prior evidence for its orientation-specific adaptation[7], though we also examined V2 and V3 for comparison (the only other regions that can be reliably localised by means of meridian mapping). All three ROIs were defined separately for each participant by functionally localising the borders between them on an inflated cortex reconstruction using meridian mapping[34]. ROIs contained only voxels responsive to the area spanned by the oriented grating stimuli (on average, the number of such voxels was 1230 (STD = 215), 1266 (STD = 211) and 1030 (STD = 198) for V1, V2 and V3 respectively). See Alink et al.[25] for more details on the stimulation protocol used during the separate fMRI localiser session.

Each grating stimulus type was presented three times during a subrun. In line with previous studies[12], repetition suppression increased across the two repetitions, with average BOLD response amplitude (% signal change) of 3.32, 2.73 and 2.68 for the 1st, 2nd and 3rd presentations respectively. To maximise repetition effects, we therefore compared responses to the first and third stimulus presentation (referred to as the initial and repeated condition). There are likely to be effects induced by intermediate presentations of the other orientation, which are modelled fully in the models described below. Any effects of the order of specific orientations were controlled by counterbalancing (i.e., averaging data over 45°-135°-45°-135°-45°-135° and 135°-45°-135°-45°-135°-45° subruns).

**fMRI data properties**. Let $B_{vtpc}$ be the BOLD signal at voxel $v$ for trial $t$ involving presentation $p$ of stimulus class $c$ (where $c$ is face or scrambled face in Experiment 1, or 45 vs. 135 degrees in Experiment 2). The number of voxels ($N_v$) was 185 for

the FFR ROI in Experiment 1, and varied from 775-1598 across participants ($M = 1100$, $SD = 220$) for the V1 ROI in Experiment 2. The number of trials (replications) for each stimulus class and presentation ($Nt$) corresponded to 49 events in Experiment 1, and the 8 sub-runs in Experiment 2. We identified 6 properties of the fMRI data, based on various analyses that have been reported in various fMRI studies[14,15], though never investigated simultaneously within a single study:

1-Mean Amplitude Modulation (MAM): this represents the mean, over voxels, trials and the two classes, of the difference in the fMRI response to the initial vs. repeated presentations:

$$\text{MAM} = \overline{\overline{\overline{B_{.1.}}}} - \overline{\overline{\overline{B_{.2.}}}} = \left( \sum_{v=1}^{N_v} \sum_{t=1}^{N_t} \sum_{c=1}^{2} B_{vt1c} - \sum_{v=1}^{N_v} \sum_{t=1}^{N_t} \sum_{c=1}^{2} B_{vt2c} \right) / (2 \times N_t \times N_v)$$

This is the typical univariate measure of repetition suppression[3].

2-Within Class Correlations (WC): this is the mean pairwise correlation of patterns over voxels, averaged across all trials and classes, and then contrasted for initial vs. repeated presentations:

$$\text{WC} = \left( \sum_{c=1}^{2} \sum_{i=1}^{N_t} \sum_{j>i}^{N_t} \text{cor}\left(B_{.i1c}, B_{.j1c}\right) - \sum_{c=1}^{2} \sum_{i=1}^{N_t} \sum_{j>i}^{N_t} \text{cor}\left(B_{.i2c}, B_{.j2c}\right) \right) \bigg/ (N_t \times (N_t - 1))$$

This captures how repetition makes patterns for the same class more or less similar.

3-Between Class Correlations (BC): This is the mean pairwise correlation of patterns over voxels for all trials of different classes, contrasted for initial vs. repeated presentations:

$$\text{BC} = \left( \sum_{i=1}^{N_t} \sum_{j=1}^{N_t} \text{cor}\left(B_{.i11}, B_{.j12}\right) - \sum_{i=1}^{N_t} \sum_{j=1}^{N_t} \text{cor}\left(B_{.i21}, B_{.j22}\right) \right) / (N_t \times N_t)$$

This captures how repetition makes patterns of the opposite class more or less similar.

4-Classification Performance (CP): The ability of MVPA to classify the two classes relates to the difference between Within- and Between-Class correlations:

$$\text{CP} = \text{WC} - \text{BC}$$

Note that this measure it is not redundant with the previous two features, since repetition might decrease both WC and BC, but decrease BC more, for example, such that CP increases. To confirm the relationship between CP and MVPA classification, we compared our CP results to the results of an analysis using linear Support Vector Machine classification (SVM, Bioinformatics Toolbox in Matlab 2012b) with leave-one-session-out cross-validation for the face dataset and with leave-one-subrun-out cross-validation for the grating dataset. Classification accuracy was found to be higher for initial than for repeated presentations of face stimuli (initial: 70%, repeated: 61%, $t(17) = 3.06$, $p < 0.005$) while the opposite was observed for grating stimuli (initial: 56%, repeated: 69%, $t(17) = -3,09$, $p < .005$)—which fits with our CP results.

5- Amplitude Modulation by Selectivity (AMS): This is a further breakdown of the first feature above, where the degree of amplitude modulation is related to the degree of "selectivity" of each voxel. Thus voxels were first binned (into six bins) by the absolute t-value of the difference between mean activity to each class (averaging across trials and both presentations, to avoid regression-to-the-mean), and then the slope estimated of a linear regression of repetition-related modulation against selectivity bin:

$$\text{AMS} = \text{slope}(\text{MAM}_b, \text{bin}_b(|t\text{test}(B_{v..1}, B_{v..2})|))$$

where slope() is the slope of best-fitting linear function, $\text{bin}_b(t)$ bins voxels to the six bins according to ascending values of $t$ from a $t$-test at each voxel across all trials of each condition, and $\text{MAM}_b$ is the amplitude of the repetition effect averaged across all voxels in bin $b$. A negative slope indicates that adaptation suppresses non-selective voxels more than selective ones.

6- AMA: This is identical to AMS above, except that voxels were binned by amplitude (averaging across trials, classes and presentations) rather than by selectivity:

$$\text{AMA} = \text{slope}\left(\text{MAM}_b, \text{bin}_b\left(\overline{\overline{B_{v...}}}\right)\right)$$

A positive slope means that adaptation suppresses more responsive voxels more, which Weiner et al.[14] claimed is indicative of scaling models.

**Modelling approach.** Assuming that neural populations (e.g. orientation columns) have a unimodal tuning-curve along a relevant stimulus dimension (e.g., orientation), at least four basic neural mechanisms of adaptation have been suggested: (1) scaling, where neural populations reduce their firing rate, i.e., their tuning curves are suppressed[3,10,16,17], (2) sharpening, where the width of neural tuning curves decreases[10], (3) repulsive shifting, where the peaks of tuning curves shift away from the adaptor[19] and (4) attractive shifting, where the peaks shift towards the adaptor[20]. These four mechanisms can be further parametrised according to whether the adaptation is (1) global, affecting all neural populations regardless of their preferred stimulus, (2) local, where adaptation is greater for neural populations whose preferences are closer to the adaptor, and (3) remote, where adaptation is greater for neural populations whose preferences are further from the adaptor. This results in a space of twelve possible models, as defined formally below.

Simulating neural responses. For simplicity, neuronal tuning curves were assumed to lie on a single stimulus dimension. These tuning curves where characterised by the firing rate, $f_i(j)$, for the $i$-th neural population in response to (the first presentation of) stimulus $j$:

$$f_i(j) = g\left(x_j; \mu_i, \sigma\right) \qquad (1)$$

where $g\left(x_j; \mu_i, \sigma\right)$ is a function of the value of stimulus $j$, $x_j$, parametrised by the preference of the $i$-th neural population (peak of its tuning curve), $\mu_i$, and the width of the tuning curve, $\sigma$. The value of $x_j$ was bounded from $0...X$ and for comparability with the circular dimension of orientation for Experiment 2 (see below), $X = \pi$. The preferred stimulus for each neural population ($\mu_i$) was selected randomly from a uniform distribution across this range, whereas the tuning width was assumed equal for all populations.

For Experiment 1, the single dimension represented faces, and the tuning curve $g\left(\theta_j; \mu_i, \sigma\right)$ was a Gaussian function (such that $\mu_i$ was its mean and $\sigma$ was its standard deviation). For Experiment 2, the stimulus dimension represented the orientation of gratings, which is a circular dimension, and so tuning curves were modelled by a von Mises distribution[16,17]. Given that the response of orientation columns is symmetrical around $180°$, $x_j$ varied from $0...\pi$ radians, such that the von Mises distribution was defined as $g\left(\theta_j; \mu_i, \sigma\right) = VM\left(2x_j; 2\mu_i, 1/\sigma\right)$. Both distributions were normalised to have a peak height of 1 (i.e., the tuning curves do not represent probability distributions).

As in Weiner et al.[14], the extent of repetition suppression was expressed through the variable $0 < c < 1$. Thus, according to the four basic neural mechanisms of adaptation, the firing rate in response to second presentation of stimulus $j$ was:

I. Scaling models: $f_i(j) = c(i,j) \times g\left(x_j; \mu_i, \sigma\right)$

II. Sharpening models: $f_i(j) = \widehat{g}\left(x_j; \mu_i, c(i,j) \times \sigma\right)$

III. Repulsive Shifting models: $f_i(j) = \widehat{g}\left(x_j; \mu_i + c'(i,j) \times \frac{X}{2}, \sigma\right)$

IV. Attractive Shifting models: $f_i(j) = \widehat{g}\left(x_j; \mu_i - c'(i,j) \times \frac{X}{2}, \sigma\right)$

where $c'(i, j)$, for shifting models, is defined below, and where $\widehat{g}$, for all models except scaling, is a re-normalised version of $g$ such that its peak height remains 1 (i.e., we assumed that sharpening and shifting do not affect the peak firing rate).

Unlike Weiner et al.[14], $c$ was itself a function of the distance between the neural preference and stimulus value, i.e., $c(i, j) = h(d(i, j); a, b)$ where $d(i, j) = \mu_i - x_j$ and $a$ and $b$ are free parameters that control the domain over which neural adaptation applied. (Note that, for the circular dimension in Experiment 2, the distance function is also circularised to $d(i, j) = \min(|d(i, j)|, X - |d(i, j)|)$). The parameter $0 < a < 1$ controlled the maximal adaptation, while $0 < b < X/2$ controlled how rapidly adaptation changed with the distance between neural preference and stimulus property. Three different distance functions were considered:

A. Global adaptation: $c(i, j) = a$

B. Local adaptation: $c(i, j) = \min\left(1, a + \left|\frac{d(i,j)}{b}\right|(1 - a)\right)$

C. Remote adaptation: $c(i, j) = \max\left(a, 1 - \left|\frac{d(i,j)}{b}\right|(1 - a)\right)$.

The parameter $b$ represents a linear slope, and in combination with the min/max operations, provides a piecewise linear function that implements the simplest form of a nonlinear saturation, as shown in Supplementary Figure 2. This nonlinearity is important for Experiment 2, to break the symmetry of results after adapting to two opposite orientations (otherwise the response of neural populations whose preference is half-way between the two adaptors could never exceed that of neural populations whose preference matches either adaptor). Global adaptation is a special case of Remote adaptation when $b = 0$, and a special case of Local adaptation when $b = \infty$.

Finally, for the two shifting models, the adaptation factor $c'(i, j)$ additionally (1) depended on the sign of the difference between neural preference and stimulus value:, i.e., $c'(i, j) = \text{sign}(d(i, j)) \times c(i, j)$, and (2) was defined such that $c'(i = j) = 0$, which means that even for local shifting, the population whose preferred stimulus matches the adaptor is not affected by adaptation. This is what is found empirically[19,35] and would actually correspond to a quadratic distance function. Rather than parametrise this quadratic function further, by defining $c'(i = j) = 0$, we are effectively limiting this quadratic function to the level of discretization of stimulus angles ($\pi/8$ here).

For multiple presentations of stimuli, we simply multiply the effects of $c$. Thus for the $p$-th presentation of a stimulus ($p > 1$):

$$c(i, j, p) = \prod_{q=2}^{q=p} c(i, j(q))$$

where $j(q)$ represents the sequence of stimuli presented. For Experiment 1, each stimulus was only repeated once, so $p = 2$. For Experiment 2, the two types of stimuli alternated across 6 blocks, such that $j = [\pi/4, 3\pi/4, \pi/4, 3\pi/4, \pi/4, 3\pi/4]$ or $j = [3\pi/4, \pi/4, 3\pi/4, \pi/4, 3\pi/4, \pi/4]$ (with the initial stimulus type counterbalanced across subruns). Given the gap between subruns, we assumed adaptation wore-off between subruns, by resetting $f$ to Eq. (1) (an assumption that is supported by the activity patterns across the whole experiment shown in Supplementary Figure 3). Note also that the pattern of simulation results below remained unchanged if we additionally simulated repetition effects occurring within each block (i.e., affecting mean response to the first block too).

**Fatigue model (special case of local-scaling).** We also implemented a two-parameter version of the local scaling model, in which the degree of adaptation is directly proportifonal to the initial response of each neural population. This activity-dependent scaling, or "fatigue" mechanism[3,40], may occur because neurons that fire more experience a greater decline in their synaptic resources, and hence are less able to fire subsequently[22]. This model is a simplification of local scaling because the initial response acts as a proxy for the degree to which a stimulus matches the neural preference. This model can be expressed as:

$$f_i(j_2) = cG\left(\theta_{j_2}, \mu_i, \sigma\right) \quad c = 1 - aG\left(\theta_{j_1}, \mu_i, \sigma\right) \quad 0 < a \le 1$$

where $j_1/j_2$ refer to first and second presentations of stimulus $j$. In other words, if $G\left(\theta_{j_2}, \mu_i, \sigma\right) = G\left(\theta_{j_1}, \mu_i, \sigma\right) = G$, then $f_i(j_2)$ is non-linearly related to initial firing rate:

$$f_i(j_2) = (1 - aG)G = G - aG^2$$

This fatigue model has only two parameters, $a$ and $\sigma$, rather than the three used for local scaling. However, while a grid search showed that this fatigue model could simultaneously fit the data features in Experiment 1, it could not fit those in Experiment 2. In particular, it could not simultaneously produce an increased CP and a decreased AMS for any of the values of $a$ and $\sigma$ examined. Thus the greater flexibility of the three-parameter local scaling model (expressed through the distance parameter $b$) seems necessary to explain all the data-features across both datasets. Therefore, the winning local scaling model cannot simply be reduced to activity-dependent adaptation, and it is likely to result from more complex neural/synaptic processes, such as interactions between neurons.

**Simulating voxel responses.** Each voxel was assumed to contain $N$ neural populations, whose preferences were randomly selected from a uniform distribution (see below). Since the BOLD response is proportional to the neural firing rate[36,37], the voxel response was simply the average firing rate of each population within that voxel. The number of neural populations per voxel, $N$, does not have a qualitative effect on the simulation results. However, it does have a quantitative effect: When $N$ is large, the majority of the voxels will be similar to each other in their response, with very weak overall voxel biases towards particular stimulus classes. If $N$ is small however, the voxels have stronger biases, and the quantitative differences among the models become more evident. Here we used $N = 8$.

We then simulated $V = 200$ voxels, and added a small amount of independent noise to each voxel, drawn from a zero-mean Gaussian distribution with standard deviation of 0.1.

To generate voxels that vary in their selectivity and activity, the value of $\mu_i$ was sampled randomly with uniform probability from 8 possible orientations from $\theta = 0 \dots 7\pi/8$ in steps of $\pi/8$. For Experiment 2, these values therefore included two neural populations that responded optimally to one of the stimuli (tuned to orientations 45°, or $\pi/4$, and 135°, or $3\pi/4$), two highly non-selective populations (tuned to orientations 0° and 90°), and four partially-selective populations in between. This sampling allowed us to generate a sufficient variety of voxel activity and types, ranging from highly-selective voxels to partially-selective to non-selective voxels for each orientation. Even though the faces used in Experiment 1 are likely to be represented along multiple dimensions, these can be projected onto a single dimension for the present argument. Thus, a prototypical face could be considered to have value $\pi/4$ and a prototypical scrambled face could be considered to have value $3\pi/4$. We also performed additional simulations in which the sampling of neural preferences was biased towards faces (i.e, a greater probability of a preference of $\pi/4$), in order to produce the greater mean activation for faces than scrambled faces that was found empirically, and the simulation results were unchanged.

Each model had 3 free parameters: $a$, $b$ and $\sigma$ (except for the Global models where there was no $b$ parameter). We explored the predictions of the twelve models for each of the 6 data features in a grid search covering a wide range of values for the three parameters. The $a$ values ranged from 0.1 to 0.9 in steps of 0.1

to cover a wide range of maximal adaptation, while $b$ values ranged from 0.1 to $\pi/2$; in steps of 0.2 radians. For $\sigma$, values ranged from 0.1 to 1 in steps of 0.2, and then from 2 to twelve in steps of 3 to cover a wide range of tuning widths. For each model, we ran 50 simulations for each of the 648 unique combinations of these three parameters (or 81 for Global models with just two parameters). For each parameter combination, we calculated the 99% confidence interval across the 50 simulations for the value associated with each of the 6 data properties, and tested whether this was above, below or overlapped zero. Figures 4, 5 summarise the results.

The fitted data properties for local scaling (using parameter values $a = 0.8$, $b = 0.4$, $s = 0.4$ for grating dataset and values $a = 0.7$, $b = 0.2$, $s = 0.2$ for the face dataset) are shown in Supplementary Figure 4. Note that, while all the qualitative effects of repetition are reproduced successfully, the absolute values of BC correlations in our simulations are negative rather than positive.

The positive correlations of BC in the data could owe to several factors, such as correlated scanner noise or temporal drift (extrinsic scanner factors). Alternatively, there could be intrinsic factors such as neural populations within a voxel that are not selective, responding equally to both stimulus classes (i.e. flat tuning curves). Such diversity in the neural tuning curves has been reported in single cell literature[38,39]. Hence, quantitative fitting of the data would require additional assumptions and scaling parameters that are not of theoretical interest in this paper. However, as a sanity check, we confirmed that we were able to achieve a positive BC correlation by adding a proportion of neural populations with flat tuning curves that respond and adapt equally to both stimulus types (Supplementary Figure 5). It is worth noting that this addition did not change the overall conclusions of this paper, i.e. local scaling was still the wining model in both datasets even after adding this extra type of neurons to emulate correlated neural activity.

**Data availability**. The Matlab code for our simulations and the region of interest fMRI response data used in this paper can be downloaded from the Open Science Foundation project: https://osf.io/ph26y/.

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

## Acknowledgements

This work was supported by British Academy postdoctoral fellowship and a Marie Curie fellowship (753441) to A.A., a Cambridge University international scholarship and IDB merit scholarship award to H.A., and Medical Research Council programme grant (SUAG/010 RG91365) to R.N.H.

## Author contributions

The initial idea was conceived by A.A., and the model simulations were performed by H.A. and R.N.H. Experiment 1 was designed and performed by R.N.H. Experiment 2 was designed and performed by A.A. Writing of the paper was performed collaboratively by all authors.

## Additional information

**Competing interests:** The authors declare no competing interests

