## [Peer Review File · Nature Communications]

Reviewers' comments:

Reviewer #1 (Remarks to the Author):

In this paper, the authors use a forward modeling approach to evaluate the ability of a variety of models of repetition suppression (RS) to address fMRI data from two experiments, one involving repeated faces and the other involving repeated oriented gratings. They consider 4 basic models (scaling, sharpening, repulsive shifting, and attractive shifting) crossed with 3 spatial domains of effects (global, local, and remote). In evaluating these models (total of 12), they also consider a variety of empirical metrics: 1) mean amplitude modulation (MAM), 2) within-class correlation (WC), 3) between-class correlation (BC), 4) classification performance (CP), 5) amplitude modulation by class selectivity (AMS), and 6) amplitude modulation by overall response amplitude (AMA). They find that only the local scaling model was able to address the full range of metrics in both experiments. Interestingly, they also find that no single metric was sufficient to identify the best model, pointing to the importance of evaluating a range of aspects of brain activity.

I found this study to be both well conceived and well carried out, with no serious methodological or logical flaws that I was able to identify. I also think that it makes an important contribution to the literature on the mechanisms underlying repetition suppression effects. I include several comments below that I hope will aid the authors in clarifying certain points for a revision.

1) The choice of models overall seems reasonable, although attraction and repulsion have been considered mainly in the literature on adaptation in primary visual cortex. The main choice that sticks out to me as a little strange is the "remote" condition. Most mechanisms proposed thusfar involve the cells that were initially driven by the adaptor (i.e. activity-dependent and stimulus-selective changes). On what basis would changes occur to cells that weren't active during the first stimulus presentation (note the width of the blue curve to the adaptor in Figure 1) ? The global condition is somewhat more reasonable given the potential for more global neuromodulatory changes to alter local tuning curves through the relative balance of excitation and inhibition. All that said, I see no real problem with including the remote condition, since at worst, it just won't perform all that well relative to the other models. The authors might comment, though, on why they included this condition.

2) The task subjects are performing in Experiment 1 is described as symmetry judgement (in the methods), but none of the face or scrambled stimuli shown in Figure 2 seem that symmetrical. Could the authors please clarify? Is it more like face/non-face judgement?

3) The authors comment on p.7 that perhaps the reason that improved classification occurs in Experiment 2 after repetition is that the timing of the gratings is more predictable. Given that the scaling model doesn't know anything about prediction, is that still your position? One might speculate instead that the sigma parameter differs slightly between representations in V1 and the fusiform gyrus (I found it notable in the Supplementary Table that one might only need to adjust the sigma parameter from 0.2 to 0.4, keeping the other parameters the same, to reverse the AMS slope from positive to negative and CP from a decrease to an increase).

4) On p.26, the authors list the SNR of the simulated voxel time series as 10. However, SNR is usually calculated in terms of a ratio of variances (or power) .. not a ratio of standard deviations (or amplitudes). The typical SNR would instead be ~ 100 . I don't necessarily think this is much of a problem. The data the authors consider are beta weights, which theoretically aren't supposed to be influenced much by noise in the dependent variable (i.e. the voxel time series). I would expect that adding more noise to the simulations would simply require more repetitions to reach stable estimates.

Reviewer #2 (Remarks to the Author):

In this paper, the authors used modeling and fMRI data in order to test four models of neural mechanisms underlying repetition suppression. This is an important and timely topic. Based on their data and analyses, they conclude that the scaling model best explains their data.

While the study of repetition suppression is important, there are several fundamental problems with the data and analyses – a combination of which makes the paper unsuitable for publication in its present form. Specifically, the paper suffers basic design and analysis flaws including circular analyses of fMRI data, inadequate experimental design to test the theoretical questions, and misrepresentation as well as a lack of scholarly knowledge regarding prior relevant studies.

Major concerns:

1) The authors perform circular analysis of their fMRI data – which has been shown to be flawed and inflate the results (see Vul et al 2009). Specifically, in Experiment 1 they use the same stimuli to define the face-ROI and to analyze the repetition effects. This is biased to show enhanced responses to non repeated images. On pages 16-17, the authors write: 'We extracted fMRI timeseries data from voxels within a combined left and right Fusiform Face Area (FFA) mask, as defined by the group univariate contrast of unfamiliar faces versus scrambled faces (averaged across initial and repeated presentations), thresholded at $p < .05$ 17 family-wise error corrected using random field theory.' However, earlier on page 16, the authors write, 'Half of the faces were famous, but only the remaining unfamiliar faces were analysed here.' Thus, if unfamiliar faces were used to define the region and the same unfamiliar faces were used to examine their experimental manipulation of stimulus repetition, the responses are non-independent and statistically flawed. In order to publish these data, the authors would need to run a separate localizer experiment to independently define the regions of interest, and then extract the fMRI signals from the present experiment to test repetition effects.

2) The data from Experiment 1 is inadequate for testing the effects of repetition on distributed responses or selectivity because they have only two conditions faces and scrambled faces. In order to adequately test the models, the authors need to use multiple types of image categories to address this as was done previously in monkey neurophysiology (McMahon and Olson, 2008; Vogels, 2016 for review) and human neuroimaging (Weiner et al., 2010) experiments. Therefore, they need to collect new data using multiple types of stimuli in experiments that were designed to address the question as opposed to using previously published data that happen to have repeated stimuli (see point 4 below).

3) The definition of the face-selective regions is not up to field standards as they are not using a specific contrast to compare responses to faces compared to many other categories. The problem with their contrast, faces > scrambled faces, is that it will also activate object-selective regions like LOC that respond similarly to faces and objects, but not to the scrambled objects. Thus, in order to define face-selective regions, they need to run an independent localizer experiment with many categories and then compare fMRI responses to faces compared to a variety of other intact and meaningful stimuli.

4) The experimental paradigm for V1 is inadequate for measuring adaptation in V1. In contrast to higher order areas, in order to generate adaptation in V1 one needs to show the same stimulus (e.g. oriented grating) for either a prolonged time or repeat it many times as well as top-it up (e.g. Fang et al 2005). However, the present study uses an experiment originally published by Alink et al. (2013) which was designed to examine orientation decoding, not adaptation. Consequently, it does not have extended presentations of the same oriented stimulus and there is no top-up adaptation. Therefore, in order to be able to present convincing evidence for adaptation mechanisms in V1 they need to collect

a new data set using paradigms as described above. These fMRI designs are modeled after neurophysiology experiments in monkeys and specifically control for the possibility that adaptation at the fMRI level in V1 could come from neuronal responses that are not from the preferred orientation – a possibility that is not controlled for in the present experiment because it was not designed to measure adaptation in V1, it was designed to measure decoding in V1.

5) The authors misattribute the results and modeling work by Weiner and colleagues. First, the authors frame this paper as the first study to do both univariate and multivariate analyses of repetition effects. This is inaccurate. Weiner et al (2010) already showed the effects of repetition on both univariate and multivariate responses in the ventral stream. Second, the study by Weiner et al was more extensive than the present study as they examined repetition effects not only within face-selective regions in the fusiform gyrus as done here, but also in other category-selective regions as well as across ventral temporal cortex more broadly. Third, the fMRI data within that paper present the distribution of adaptation effects, as well as the slope fitting repeated vs non-repeated stimuli across voxels, both of which are multivoxel analyses. This slope was also calculated across categories because the theoretical models make different predictions regarding the repeated/nonrepeated ratio across categories. Fourth, the modeling in Weiner et al is also multivariate because there are multiple measures per simulated neuron, and simulated voxels. This error should be corrected in the present manuscript as they are not the first to model multivariate responses.

6) The scholarly knowledge of the paper is brought to question as the authors do not cite any of crucial work from the Vogels lab. In terms of examining the scaling, sharpening, and facilitation models in monkey inferotemporal cortex, it could be said that Vogels and colleagues have explored this space and underlying theoretical mechanisms more than any other lab. Specifically, while the present authors conclude that not just local scaling can explain repetition suppression effects, De Baene and Vogels (2010, Cereb Cortex) used single-cell and LFP measures in macaque IT with formal models and already concluded the same thing.

7) The authors do little to justify the selection of the 6 parameters used to optimize their model. Though the authors note that these 6 parameters are found in some studies in the literature, they do not explain why these parameters are all necessary (and sufficient) to adequately model fMRI adaptation. Moreover, the parameters are not independent of one another and give rise to problems of over-fitting as some features are represented in multiple components of the models.

8) The authors argue that only local scaling could fit all 6 parameters across the two brain locations. The authors do not justify why mechanisms of adaptation should be the same in varied brain regions. In fact, a substantial body of work in both monkeys and humans indicates that adaptation mechanisms are different across brain regions.

Reviewer #3 (Remarks to the Author):

Alink and colleagues investigate potential mechanisms of repetition suppression using data from two separate fMRI experiments. They find that only one model, local scaling, is consistent with all the data. The underlying mechanisms of repetition suppression have been debated over many years and there are many things to like about this manuscript, in particular the testing of multiple computational models using multiple datasets with different stimuli and focusing on different brain areas. This sets the study apart from much prior work and I think the manuscript has the potential to have a large impact.

Nevertheless, I think the manuscript would benefit from some substantial revisions. I was particularly

concerned that most of the key details are only evident from the supplementary material and the main text reads quite superficially as a consequence. It also makes the manuscript hard to follow if you really want to evaluate the analyses. I understand the authors are trying to make the manuscript more accessible, but I think they are doing themselves a disservice by not elaborating on the analytical details in the main part of the manuscript. And I think it is also important that they justify some of the analytical decisions they make and give more motivation. As it stands, I'm confused about some of the details and am not sure the authors' approach is the most sensible.

Below I list some comments and questions that I think the authors could address to improve the manuscript (in no particular order).

1) How much of a limitation is it that the authors only tested two stimulus types in each experiment? They tested faces and scrambled faces in Experiment 1 and two orthogonal gratings in Experiment 2. Doesn't this make it hard to establish how tuning functions are changing?

2) I think it would be good to show the data separately for each stimulus type rather than averaging across the stimuli as I assume is currently done. This might not matter much for the grating stimuli where preferences will vary from voxel to voxel, but in FFA all voxels presumably show a stronger response to the faces and it would be helpful to see how the effects plotted in Figure 3, for example, vary by stimulus.

3) It's not completely clear to me why the authors are not looking at the delayed repeats in Experiment 1. Wouldn't this allow a greater test of the generalization of the findings, especially given the prior work from Weiner and colleagues?

4) It's not completely clear to me how the visualizations in Figures 4 and 5 were produced. Maybe I'm missing some of the detail, but it's critical the authors describe these plots in sufficient detail in the main manuscript – these are the key figures showing the fit between the models and the data. What exactly do the different colors reflect? According to the supplementary methods, the authors ran 50 simulations for each of the 8000 parameter combinations and then computed 99% confidence intervals for each set of 50 simulations. So, doesn't that mean there are 8000 sets of confidence intervals that are then converted to the color scale for each circle in Figure 4. How was this done? Or do the authors mean that the confidence intervals were calculated across all 50 X 8000 simulations. Please clarify.

5) I'm somewhat concerned the authors are throwing out information that may be informative. As I understand it, a set of parameters for which all but one simulation cross zero and another set of parameters for which 50% cross zero will be counted as the same. Please discuss and clarify. How would the results change, if at all, if the authors considered 95% confidence intervals rather than 99%. Is it essentially impossible to have 'flat' as the authors list in the figure legend?

6) As the authors note, the generalizability of their findings is unclear and could be affected by many factors. It strikes me that the authors could address some of these issues by extending their current testing. I noted above the issue of the delayed repeats, which are present in one of their datasets. Further, the authors could test more brain regions e.g. the occipital face area for Experiment 1, V2/V3 for Experiment 2. Do the authors findings generalize across brain regions within a given experiment?

7) Throughout both the main part of the manuscript and the supplementary material, the figure legends need much more detail to make the figures comprehensible.

8) The justification for only considering first versus third repeats of the orientations in Experiment 2 is not completely clear to me. Please expand and clarify.

Overall, I really like the approach of this study, I think it is important and could have large impact – I enjoyed reading it. But I need more details and more clarity in the exposition to have full confidence in the findings.

Point-by-Point reply to Three Reviewers

Reviewer #1 (Remarks to the Author):

In this paper, the authors use a forward modeling approach to evaluate the ability of a variety of models of repetition suppression (RS) to address fMRI data from two experiments, one involving repeated faces and the other involving repeated oriented gratings. They consider 4 basic models (scaling, sharpening, repulsive shifting, and attractive shifting) crossed with 3 spatial domains of effects (global, local, and remote). In evaluating these models (total of 12), they also consider a variety of empirical metrics: 1) mean amplitude modulation (MAM), 2) within-class correlation (WC), 3) between-class correlation (BC), 4) classification performance (CP), 5) amplitude modulation by class selectivity (AMS), and 6) amplitude modulation by overall response amplitude (AMA). They find that only the local scaling model was able to address the full range of metrics in both experiments. Interestingly, they also find that no single metric was sufficient to identify the best model, pointing to the importance of evaluating a range of aspects of brain activity.

I found this study to be both well conceived and well carried out, with no serious methodological or logical flaws that I was able to identify. I also think that it makes an important contribution to the literature on the mechanisms underlying repetition suppression effects. I include several comments below that I hope will aid the authors in clarifying certain points for a revision.

1) The choice of models overall seems reasonable, although attraction and repulsion have been considered mainly in the literature on adaptation in primary visual cortex. The main choice that sticks out to me as a little strange is the "remote" condition. Most mechanisms proposed thus far involve the cells that were initially driven by the adaptor (i.e. activity-dependent and stimulus-selective changes). On what basis would changes occur to cells that weren't active during the first stimulus presentation (note the width of the blue curve to the adaptor in Figure 1) ? The global condition is somewhat more reasonable given the potential for more global neuromodulatory changes to alter local tuning curves through the relative balance of excitation and inhibition. All that said, I see no real problem with including the remote condition, since at worst, it just won't perform all that well relative to the other models. The authors might comment, though, on why they included this condition.

We fully agree that local or global effects are more easily understood. However, remote effects could arise through interactions between neural populations within a more complex circuit. For example "remote scaling", whereby neurons whose preference is furthest from the stimulus are suppressed most, could be implemented by strengthening of inhibitory interactions between neurons. Indeed, precisely this mechanism of remote scaling was proposed by Wiggs & Martin (1998) to explain repetition effects, whereby neurons less sensitive to a stimulus "drop out" of the representation of that stimulus. We have added a sentence in the Introduction to expand on this point, and thank the reviewer for prompting this:

"Global effects could arise, for example, from neuromodulatory changes that affect a whole brain region; local effects could arise from activity-dependent changes such as synaptic depression²¹; while remote effects could arise from strengthening of inhibitory interneurons that implement "winner-takes-all" dynamics²²."

2) The task subjects are performing in Experiment 1 is described as symmetry judgement (in the methods), but none of the face or scrambled stimuli shown in Figure 2 seem that symmetrical. Could the authors please clarify? Is it more like face/non-face judgement?

We apologise for not being clear. Participants were asked to judge the left-right symmetry of the image about a vertical line through the centre of the image. They were told that no image is perfectly symmetrical, but they should respond according to their perceived degree of symmetry. They were given some practice at this task, using same type of stimuli as used in the main experiment (but not same images), so that they could establish the range of likely symmetries. We now explain this more clearly in the Methods.

3) The authors comment on p.7 that perhaps the reason that improved classification occurs in Experiment 2 after repetition is that the timing of the gratings is more predictable. Given that the scaling model doesn't know anything about prediction, is that still your position? One might speculate instead that the sigma parameter differs slightly between representations in V1 and the fusiform gyrus (I found it notable in the

Supplementary Table that one might only need to adjust the sigma parameter from 0.2 to 0.4, keeping the other parameters the same, to reverse the AMS slope from positive to negative and CP from a decrease to an increase).

This is a very perceptive comment. It is indeed the sigma parameter that is critical for the local scaling model to produce different patterns for CP and AMS across Experiment 1 and Experiment 2. We had investigated the reasons, but did not think there was space in our original submission to cover in detail. We now have added the following text to the Discussion:

“How is the local scaling model flexible enough to produce opposite effects of repetition on CP and AMS across Experiment 1 and 2? The critical parameter turns out to be σ , the width of the neuronal tuning curves. As can be seen in the Supplementary Table, the optimal σ values ranged between 0.2 and 0.4 for Experiment 1, but between 0.4 and 1 for Experiment 2. When σ is low, a greater number of voxels have a selectivity for one stimulus (by chance), and so when these are suppressed, there is a decrease in WC after repetition (because highly tuned voxels are suppressed more), but relatively less decrease in BC. On the other hand, when σ is large, there are a fewer selective voxels and hence these are suppressed less, and there is less reduction in WC after repetition and a relatively larger reduction in BC. This trade-off between WC and BC allows CP to decrease for the face-scrambled distinction in Experiment 1 (when σ is low) but increase for the orientation distinction in Experiment 2 (when σ is high). For AMS, when σ is low and a greater number of voxels are highly selective, the effect of local scaling is to lower their selectivity ranking. Because such selective voxels also show more suppression, there is a positive dependency between suppression and selectivity. Conversely, when σ is large, local scaling tends to increase the selectivity ranking of less selective neurons, so there is a negative dependency between suppression and selectivity.”

Note that it was not our intention to suggest that it is the more predictable timing of the gratings that causes the different results across experiments. While their greater predictability may be a factor, we think it is not as likely as the differences in the ROI and type of stimuli, as we now explain (continuing above paragraph):

“Note that σ reflects the tuning curve width relative to the range of possible stimulus values (which was fixed as X here for both experiments), so the differences between experiments could be a property of the different ROIs and/or the different stimuli used (the differences in CP and AMS could also reflect other procedural differences between the experiments, but it is less obvious how such differences would affect σ). In any case, it is interesting that a single parameter within a simple model can produce such a range of different qualitative outcomes at the level of fMRI analysis, again questioning any “inverse” inference one might be tempted to make from fMRI.”

4) On p.26, the authors list the SNR of the simulated voxel time series as 10. However, SNR is usually calculated in terms of a ratio of variances (or power) .. not a ratio of standard deviations (or amplitudes). The typical SNR would instead be ~ 100. I don't necessarily think this is much of a problem. The data the authors consider are beta weights, which theoretically aren't supposed to be influenced much by noise in the dependent variable (i.e. the voxel time series). I would expect that adding more noise to the simulations would simply require more repetitions to reach stable estimates.

We apologize and have corrected this value to 100 to be more conventional.

Reviewer #2 (Remarks to the Author):

In this paper, the authors used modeling and fMRI data in order to test four models of neural mechanisms underlying repetition suppression. This is an important and timely topic. Based on their data and analyses, they conclude that the scaling model best explains their data.

While the study of repetition suppression is important, there are several fundamental problems with the data and analyses – a combination of which makes the paper unsuitable for publication in its present form. Specifically, the paper suffers basic design and analysis flaws including circular analyses of fMRI data, inadequate experimental design to test the theoretical questions, and misrepresentation as well as a lack of scholarly knowledge regarding prior relevant studies.

Major concerns:

1) The authors perform circular analysis of their fMRI data – which has been shown to be flawed and inflate the results (see Vul et al 2009). Specifically, in Experiment 1 they use the same stimuli to define the face-ROI and to analyze the repetition effects. This is biased to show enhanced responses to non repeated images. On pages 16-17, the authors write: 'We extracted fMRI timeseries data from voxels within a combined left and right Fusiform Face Area (FFA) mask, as defined by the group univariate contrast of unfamiliar faces versus scrambled faces (averaged across initial and repeated presentations), thresholded at $p < .05$ 17 family-wise error corrected using random field theory.' However, earlier on page 16, the authors write, 'Half of the faces were famous, but only the remaining unfamiliar faces were analysed here.' Thus, if unfamiliar faces were used to define the region and the same unfamiliar faces were used to examine their experimental manipulation of stimulus repetition, the responses are non-independent and statistically flawed. In order to publish these data, the authors would need to run a separate localizer experiment to independently define the regions of interest, and then extract the fMRI signals from the present experiment to test repetition effects.

This is a very serious accusation and we fully understand the editor's decision to reject a manuscript based on this comment. The accusation, however, is completely wrong. We did use the same stimuli to define the ROI and to analyze repetition effects. However, the contrast used to define the ROI was orthogonal to the contrast used to test repetition effects. As we clearly stated, the contrast to define the ROI was averaged over initial and repeated presentations. This means *there is no bias to show enhanced responses to non-repeated images* (or reduced responses for that matter). There is no circularity.

In brief, the reason for this lack of bias is that the sum of two independent numbers is independent of their difference, despite the fact they are mixtures of the same data. More formally, if we order four conditions as [Faces_Initial Scrambled_Initial Faces_Repeat Scrambled_Repeat] then in statistical terms, the contrast weights (**c**) for the ROI definition are [1 1 -1 -1], while those for the repetition effects are [1 -1 1 -1] (or [1 -1 0 0] or [0 0 1 -1]), which are orthogonal vectors. These weights were evaluated within the context of a "group-level" General Linear Model with a design matrix (**X**) capturing each participant's mean response per condition, in which the orthogonal regressors (columns of **X**) mean that the contrasts themselves (the weights multiplied by the design matrix, i.e. **cX**) are also necessarily orthogonal.

This point about defining ROIs from orthogonal contrasts has been discussed at length by Friston et al (2006, Neuroimage, 30, 1077-1087; see also Friston & Henson, Neuroimage, 30, 1097-1099). Indeed, that paper listed the advantages of defining ROIs from the same data rather than from independent localizers, where changes in task, stimuli and context might influence the functional response. We do not want to rehearse this argument further, since there are other occasions (such as the retinotopic mapping we used to define V1 in our second experiment) when separate localizers are necessary, but we cite these papers to point out that peer-reviewed publications have made the same point that we make here.

2) The data from Experiment 1 is inadequate for testing the effects of repetition on distributed responses or selectivity because they have only two conditions faces and scrambled faces. In order to adequately test the models, the authors need to use multiple types of image categories to address this as was done previously in monkey neurophysiology (McMahon and Olson, 2008; Vogels, 2016 for review) and human neuroimaging (Weiner et al., 2010) experiments. Therefore, they need to collect new data using multiple types of stimuli in experiments that were designed to address the question as opposed to using previously published data that happen to have repeated stimuli (see point 4 below).

Points 2 and 4 are related as they both question the adequacy of the paradigms used in both Experiment 1 and Experiment 2. We start by noting that this judgment strongly contrasts with that of Reviewer 1 “I found this study to be both well-conceived and well carried out, with no serious methodological or logical flaws that I was able to identify”, and Reviewer 3 did not raise any substantial issues related to our experimental paradigm.

In their Point 2, Reviewer 2 claims that “data from Experiment 1 is inadequate for testing the effects of repetition on distributed responses or selectivity because they have only two conditions faces and scrambled faces.” However, Reviewer 2 provides no logical reason for why more than two stimulus categories are needed. It is true that using a range of stimuli ordered along a relevant dimension would enable plotting of voxel “tuning curves”, but the point of our modelling is to show that such voxel-level data patterns cannot be used to directly infer neuronal-level tuning curves. Moreover, our modelling work shows that two stimulus categories *are* sufficient to distinguish neuronal sharpening from neuronal scaling, so the Reviewer is simply incorrect that our designs are insufficient: we show clearly that local scaling is the only model that can fit all our data. In brief, the Reviewer’s claim that more than two conditions is needed is unsubstantiated, whereas our modelling results substantiate our claim that two conditions are sufficient to uniquely identify one of the twelve plausible neural mechanisms that we considered.

3) The definition of the face-selective regions is not up to field standards as they are not using a specific contrast to compare responses to faces compared to many other categories. The problem with their contrast, faces> scrambled faces, is that it will also activate object-selective regions like LOC that respond similarly to faces and objects, but not to the scrambled objects. Thus, in order to define face-selective regions, they need to run an independent localizer experiment with many categories and then compare fMRI responses to faces compared to a variety of other intact and meaningful stimuli.

We would contest that there is only one way to define FFA in terms of the stimuli contrasted: whether the best comparison category is other non-face objects (suggested by Reviewer) or phase-scrambled faces (as we used) depends on one’s theory about what drives FFA. While non-face objects do control for higher-level concepts like meaningfulness, they do not control for low-level visual properties. Our phase-scrambled faces, on the other hand, are designed to have the same spatial power spectrum, which controls for some of the main (admittedly not all) differences earlier in the visual stream. Having said this, we do not claim to know the important dimensions to control, so would be happy not to use the phrase FFA, or “face-selective”, if that implies certain conventions to other researchers, and could use instead “face-responsive”. More importantly, the FFA (and OFA) regions we identified overlap with published results from many other definitions and seriously doubt that any small differences would affect any of the conclusions in our paper.

4) The experimental paradigm for V1 is inadequate for measuring adaptation in V1. In contrast to higher order areas, in order to generate adaptation in V1 one needs to show the same stimulus (e.g. oriented grating) for either a prolonged time or repeat it many times as well as top-it up (e.g. Fang et al 2005). However, the present study uses an experiment originally published by Alink et al. (2013) which was designed to examine orientation decoding, not adaptation. Consequently, it does not have extended presentations of the same oriented stimulus and there is no top-up adaptation. Therefore, in order to be able to present convincing evidence for adaptation mechanisms in V1 they need to collect a new data set using paradigms as described above. These fMRI designs are modeled after neurophysiology experiments in monkeys and specifically control for the possibility that adaptation at the fMRI level in V1 could come from neuronal responses that are not from the preferred orientation – a possibility that is not controlled for in the present experiment because it was not designed to measure adaptation in V1, it was designed to measure decoding in V1.

We contest the claim that the paradigm of Experiment 2 is inadequate for measuring adaptation in V1 because we did not “show the same stimulus (e.g. oriented grating) for either a prolonged time or repeat it many times as well as top-it up (e.g. Fang et al 2005)”. Firstly, this statement is simply empirically wrong, since we did observe fMRI response amplitude decrease for repeated grating stimuli ($p < .00001$). Secondly, Reviewer 2 seems to have missed the fact that each stimulus block entailed many repetitions of iso-oriented gratings (page 17: “Each block lasted 14s and contained 28 phase-randomized gratings of one orientation, presented at a frequency of 2 Hz”), i.e, we did “repeat it many times”. The issue about top-up of adaptation is not relevant to our design, since we measure the mean response across alternating blocks, rather than to a single probe stimulus, and in any case, a recent study by Weigelt and colleagues (2008) showed that orientation fMRI adaptation in V1 can be observed using experimental designs without long pre-adaptation stimulus periods and top-up stimuli.

5) The authors misattribute the results and modeling work by Weiner and colleagues. First, the authors frame this paper as the first study to do both univariate and multivariate analyses of repetition effects. This is

inaccurate. Weiner et al (2010) already showed the effects of repetition on both univariate and multivariate responses in the ventral stream. Second, the study by Weiner et al was more extensive than the present study as they examined repetition effects not only within face-selective regions in the fusiform gyrus as done here, but also in other category-selective regions as well as across ventral temporal cortex more broadly. Third, the fMRI data within that paper present the distribution of adaptation effects, as well as the slope fitting repeated vs non-repeated stimuli across voxels, both of which are multivoxel analyses. This slope was also calculated across categories because the theoretical models make different predictions regarding the repeated/nonrepeated ratio across categories. Fourth, the modeling in Weiner et al is also multivariate because there are multiple measures per simulated neuron, and simulated voxels. This error should be corrected in the present manuscript as they are not the first to model multivariate responses.

We did not explicitly frame our paper as being the first study to do both univariate and multivariate analyses of repetition effects. There is only one sentence in our manuscript that could be taken as implicitly suggesting this: "To our knowledge, only Kok and colleagues¹⁵ have previously examined a combination of univariate and multivariate fMRI data features to elucidate neural mechanisms of fMRI response reductions." Thus in this sentence, we are in fact crediting another study with examining the relationship between fMRI patterns and fMRI response reductions. The reviewer is correct in pointing out that some of the subsidiary analyses by Weiner et al. were multivariate, and we have corrected this oversight by changing this sentence "To our knowledge, only two studies^{14,15} have previously examined a combination of univariate and multivariate fMRI data features to elucidate neural mechanisms of fMRI response reductions." The fact remains that no previous study formally explored the range of neural models in their ability to simultaneously fit the range of empirical findings, and thereby able to infer a single neural mechanism of local scaling (where the scaling varies according to difference between neuronal preference and stimulus attribute; a property not considered by previous studies).

6) The scholarly knowledge of the paper is brought to question as the authors do not cite any of crucial work from the Vogels lab. In terms of examining the scaling, sharpening, and facilitation models in monkey inferotemporal cortex, it could be said that Vogels and colleagues have explored this space and underlying theoretical mechanisms more than any other lab. Specifically, while the present authors conclude that not just local scaling can explain repetition suppression effects, De Baene and Vogels (2010, Cereb Cortex) used single-cell and LFP measures in macaque IT with formal models and already concluded the same thing.

We have addressed this concern in the revised version of manuscript by citing the most relevant paper from the Vogels lab in the first paragraph of the discussion.

7) The authors do little to justify the selection of the 6 parameters used to optimize their model. Though the authors note that these 6 parameters are found in some studies in the literature, they do not explain why these parameters are all necessary (and sufficient) to adequately model fMRI adaptation. Moreover, the parameters are not independent of one another and give rise to problems of over-fitting as some features are represented in multiple components of the models.

This question suggests some confusion. Our models did not have 6 parameters – they had between 2-3 parameters (a, b and sigma), which are clearly stated and defined on many occasions throughout the paper. Furthermore, we did not optimize these parameters: we performed a grid search over a range of plausible values, and showed that only one model could fit the data using parameter values within a subspace of this range (in a sense, we were integrating over parameter values).

8) The authors argue that only local scaling could fit all 6 parameters across the two brain locations. The authors do not justify why mechanisms of adaptation should be the same in varied brain regions. In fact, a substantial body of work in both monkeys and humans indicates that adaptation mechanisms are different across brain regions.

In the revised manuscript, prompted by Reviewer 2 and 3, we report responses from several other brain regions (OFA, V2 and V3), which all converge with our findings in FFA and V1, suggesting a common local scaling mechanism; see Supplementary Figures 6-7.

Reviewer #3 (Remarks to the Author):

Alink and colleagues investigate potential mechanisms of repetition suppression using data from two separate fMRI experiments. They find that only one model, local scaling, is consistent with all the data. The underlying mechanisms of repetition suppression have been debated over many years and there are many things to like about this manuscript, in particular the testing of multiple computational models using multiple datasets with different stimuli and focusing on different brain areas. This sets the study apart from much prior work and I think the manuscript has the potential to have a large impact.

Nevertheless, I think the manuscript would benefit from some substantial revisions. I was particularly concerned that most of the key details are only evident from the supplementary material and the main text reads quite superficially as a consequence. It also makes the manuscript hard to follow if you really want to evaluate the analyses. I understand the authors are trying to make the manuscript more accessible, but I think they are doing themselves a disservice by not elaborating on the analytical details in the main part of the manuscript. And I think it is also important that they justify some of the analytical decisions they make and give more motivation. As it stands, I'm confused about some of the details and am not sure the authors' approach is the most sensible.

Below I list some comments and questions that I think the authors could address to improve the manuscript (in no particular order).

1) How much of a limitation is it that the authors only tested two stimulus types in each experiment? They tested faces and scrambled faces in Experiment 1 and two orthogonal gratings in Experiment 2. Doesn't this make it hard to establish how tuning functions are changing?

It is true that using a range of stimuli ordered along a relevant dimension would enable plotting of voxel "tuning curves", but the point of our modelling is to show that such voxel-level data patterns cannot be used to directly infer neuronal-level tuning curves. Moreover, our modelling work shows that two stimulus categories are sufficient to distinguish different repetition effects on neural tuning curves, in that we show clearly that local scaling is the only model that can fit all our data. In other words, our modelling results substantiate our claim that two conditions are sufficient to uniquely identify one of the twelve plausible neural mechanisms that we considered.

2) I think it would be good to show the data separately for each stimulus type rather than averaging across the stimuli as I assume is currently done. This might not matter much for the grating stimuli where preferences will vary from voxel to voxel, but in FFA all voxels presumably show a stronger response the faces and it would be helpful to see how the effects plotted in Figure 3, for example, vary by stimulus.

We now include a figure (Supplementary Figure 9) showing repetition effects on MAM and AMA separately for face and scrambled face stimuli. Neither of these data features qualitatively different for the two stimulus classes. Note that we are unable to separate effects for the stimulus classes for the other four data features because they rely on between-stimulus-class comparisons.

3) It's not completely clear to me why the authors are not looking at the delayed repeats in Experiment 1. Wouldn't this allow a greater test of the generalization of the findings, especially given the prior work from Weiner and colleagues?

We now include a figure (Supplementary Figure 8) showing repetition effects on for the delayed repetitions of faces in Experiment 1. These are qualitatively similar to the effects for the immediate repetition (Figure 3), so suggest a common mechanism of local scaling (at least for these stimuli and ROI).

4) It's not completely clear to me how the visualizations in Figures 4 and 5 were produced. Maybe I'm missing some of the detail, but it's critical the authors describe these plots in sufficient detail in the main manuscript – these are the key figures showing the fit between the models and the data. What exactly do the different colors reflect? According to the supplementary methods, the authors ran 50 simulations for each of the 8000 parameter combinations and then computed 99% confidence intervals for each set of 50 simulations. So, doesn't that mean there are 8000 sets of confidence intervals that are then converted to the color scale for each circle in Figure 4. How was this done? Or do the authors mean that the confidence intervals were calculated across all 50 X 8000 simulations. Please clarify.

The confidence intervals (for differences or slopes, depending on data feature) were calculated across the 50 simulations, for each parameter combination in each model and for each data feature separately. For clarity, we now have added the following explanations of figure 4 and 5 on page 10 and 12.

p.10: "Each circle represents a specific model, data feature and experiment. If there existed at least one parameter combination in which a model's 99% confidence interval for a feature was above zero, then that circle included red. If there existed at least one parameter combination in which the confidence interval was below zero, then that circle included blue. If at least one combination produced a confidence interval that straddled zero, then that circle included white. When different parameter combinations produced two or more of these patterns, the circle was given a mixture of the corresponding colors."

p.12: "If a circle is colored green, then there existed a parameter combination in which that model's 99% confidence interval was consistent with the significant effect for that data feature (in that experiment); otherwise a circle is red. The particular parameter combination chosen was the one that simultaneously reproduced as many of the six data features as possible. Thus only if at least one parameter combination could simultaneously reproduce all six features would a whole column of Figure 5 be green."

5) I'm somewhat concerned the authors are throwing out information that may be informative. As I understand it, a set of parameters for which all but one simulation cross zero and another set of parameters for which 50% cross zero will be counted as the same. Please discuss and clarify. How would the results change, if at all, if the authors considered 95% confidence intervals rather than 99%. Is it essentially impossible to have 'flat' as the authors list in the figure legend?

We could report the range of confidence intervals across the 8000 parameter combinations, but that would make an already complex table even harder to understand. Moreover, because the absolute values of the simulated data features are somewhat arbitrary (since we are not fitting the data quantitatively, given all the arbitrary scaling in fMRI data) – we only care about the significance of qualitative trends (increases, decreases or no change) – we see little value in reporting such additional quantitative information. The advantage of binarizing with confidence intervals is to make the general flexibility of each model / informativeness of each data feature in Figure 4 (and accuracy of each model in Figure 5) be apparent at a glance.

A set of parameters for which all but one simulation produced results one side of zero would likely include red or blue in Figure 4 (depending on sign), because that would indicate at least one parameter combination whose 99% confidence interval likely did not overlap zero (we say "likely" because the confidence interval is not exactly the same as the proportion of simulations). Whether a set of parameters in which 50% of simulations lay either side of zero would lead to a circle being colored white would depend on all the other parameter combinations: only if all of them produced confidence intervals that overlapped zero would it be colored white.

It is true that the chance of a circle being colored white in Figure 4 would decrease if we decreased the model's confidence level (since the confidence intervals would shrink). However, we only really care about situations where we can be confident that a parameter combination is different from zero, because all the data features were significantly different from zero (i.e., no features were "flat" in the data). We chose 99% as a reasonably strict scientific level of evidence, however we repeated with a 95% level and the conclusions one would draw from Figures 4 and 5 were unchanged.

6) As the authors note, the generalizability of their findings is unclear and could be affected by many factors. It strikes me that the authors could address some of these issues by extending their current testing. I noted above the issue of the delayed repeats, which are present in one of their datasets. Further, the authors could test more brain regions e.g. the occipital face area for Experiment 1, V2/V3 for Experiment 2. Do the authors findings generalize across brain regions within a given experiment?

As requested, we analysed data from OFA in Experiment 1 (which was the only other ROI surviving the corrected threshold in our functional localizer), and for V2 and V3 in Experiment 2 (which are the two other retinotopic areas that can be reliably localized by means of meridian mapping). The results are shown in Supplementary Figures 6-7. The data features are qualitatively similar across ROIs within the same experiment (while still differing across experiments), consistent with a single local scaling model applying in all ROIs.

7) Throughout both the main part of the manuscript and the supplementary material, the figure legends need much more detail to make the figures comprehensible.

We apologize, and have made the figure legends more comprehensive - explaining all terms and abbreviations.

8) The justification for only considering first versus third repeats of the orientations in Experiment 2 is not completely clear to me. Please expand and clarify.

We have expanded our justification:

“Each grating stimulus type was presented three times during a subrun. In line with previous studies¹², repetition suppression increased across the two repetitions, with average BOLD response amplitude (% signal change) of 3.32, 2.73 and 2.68 for the 1st, 2nd and 3rd presentations respectively. To maximize repetition effects, we therefore compared responses to the first and third stimulus presentation (referred to as the initial and repeated condition).”

In fact, the results are very similar when we examine the second presentation; the repetition effects are simply slightly weaker.

Overall, I really like the approach of this study, I think it is important and could have large impact – I enjoyed reading it. But I need more details and more clarity in the exposition to have full confidence in the findings.

We thank the Reviewer for their positive assessment.

REVIEWERS' COMMENTS:

Reviewer #1 (Remarks to the Author):

The authors have done an excellent job at responding to all of my previous comments, and I was already quite enthusiastic about this paper. Congratulations on a strong contribution.

Reviewer #2 (Remarks to the Author):

This is a revised version of the paper: From Neurons to voxels- repetition suppression is best modeled by local scaling

This paper implements different models of neural effects of repetition to predict fMRI responses to repeated stimuli in the fusiform gyrus to faces and scrambled face and gratings in V1 to test what best explains repetition effects with fMRI. They conclude that local scaling best explained their results

In the prior review, I have highlighted the strength of the modeling approach, but have raised several major concerns regarding the inadequacy of the experiment to resolve the different theories, the problematic voxel selection, the novelty, and the fact that data do not support the conclusions

Unfortunately, in this revised version the authors did not address any of these concerns. Instead of performing the controls suggested in the prior review to address the main concerns, the authors use rhetorical arguments why they should not perform any additional measurements or controls. This is unsatisfactory.

As they have addressed none of the major concerns, I cannot recommend publication of this paper.

I will reiterate the major concerns again

1) The experimental design is inadequate to examine the effects of repetition suppression because in each experiment they have only 2 conditions (Exp 1: faces/scrambled faces; Exp 2: 45 and 135 oriented grating). In order to make inferences as how repetition affects tuning curves (as illustrated in Figure 1) they need to include multiple stimuli in these experiments. Simply put, 2 stimuli are insufficient to determine tuning. One needs to have a range of stimuli for that as there are many possible curves that can occur between 2 points (reflecting their experimental conditions). Because they do not have data for other points, they cannot derive any inferences about tuning, selectivity, or the effect on repetition on these factors. Furthermore, several prior experiments in both monkey electrophysiology and human fMRI (e.g. from the Movshon, Kohn, Vogels, and Grill-Spector labs) have included multiple stimuli (e.g. multiple oriented gratings or motion directions or objects) to measure the effect of repetition on tuning curves. The only way to alleviate this key concern is to collect more data with more stimulus types for faces (e.g. using multiple levels of morphing from faces to scrambled faces) and more orientations for V1.

Because their data does not provide sufficient constraints to the model, their modeling work is insufficient for distinguishing between different neural accounts of repetition suppression

2) Definition of face-selective activations. The authors contrast faces to scrambled faces and call the activations on the fusiform gyrus the FFA. This is problematic for two reasons:

(i) many voxels in the fusiform gyrus and nearby regions on the occipital temporal sulcus respond more strongly to faces than scrambled faces, but that does not make them face-selective. For

example, they may respond to bodies, words, or objects more than faces even as responses for faces are higher than scrambled faces. Specifically, next to the FFA there is both a body-selective region and a word-selective region. Why is this important for this experiment? This is important because their modeling hinges on that they know what is the tuning and the preferred stimulus for their voxels. But in fact, their measurements are not guaranteed to find the selective voxels.

(ii) The fact that they are calling their activations the FFA is misleading as they do not identify the FFA as Kanwisher does (faces vs. many other objects, see her 1997 and 2017 Journal of Neuroscience papers). Thus, it is unclear whether their activations are the same cortical region that Kanwisher calls the FFA. In other words, calling two different regions by the same name is misleading and adds confusion to the field.

3) Circularity of analysis. In the prior review, I have highlighted the dangers of circular analyses by using the same voxels to select the region and perform the main analysis of interest. The authors replied by writing that (1) I am wrong that they have a bias and (2) a 2006 paper co-authored by Henson, Friston, and colleagues argued why you should never use localisers. I disagree on both accounts. First, there is always a voxel-selection bias when using the same voxels to select a region and analyze its data. Second, in the same 2006 NeuroImage issue, Saxe, Kanwisher, and colleagues argued why it is always necessary to run functional localisers. Thus, at minimum they should acknowledge the other point of view. There are many ways in which they could have run a control to address the concern raised in the prior review. They could have used ½ the data to select the voxels and ½ the data to analyse repetition effects. Alternatively, they could have used an anatomical region to select voxels. This would have alleviated my concern. Arguing with a rhetoric that is more than a decade old and is not broadly accepted does not alleviate my concerns.

4) Novelty. The authors should build on prior results rather than superficially cite them to artificially highlight the novelty of their study. It is necessary for developing knowledge in the field to accurately state what was done in prior papers and how their data relates to prior reports. For example, while they now cite some papers that were ignored in the prior version of the manuscript, these citations are cursory and do not engage with the findings or anchor their paper accurately compared to prior research.

The novelty in their study is the nice and comprehensive array of computational models. The novelty is not in that they are the first to examine both univariate and multivariate effects of repetition.

For example, the authors write: "A previous study by Weiner and colleagues(14) formally modelled the relationship between neuron-level and voxel-level repetition effects. This study was an important demonstration that neural scaling and neural sharpening can reproduce similar effects of repetition on the mean fMRI response across voxels (a "univariate" response), but did not consider the effect of repetition on patterns of fMRI responses across voxels (a "multivariate" response). Indeed, a recent study of multi-voxel patterns by Kok et al.(15) that prediction of upcoming stimuli (which may also arise from repetition) improved classification of those stimuli, and this was used to support the idea of neural sharpening. However, neither study considered how the effects of repetition on a neural population depend on the difference between a stimulus property and the preference of that neural population for that property."

This characterisation that none of the prior studies examined systematically the effect of repetition on distributed responses is inaccurate. First, Figures 4 and 6 in Weiner et al show the distribution of repetition effects across voxels. Second, Figure 8 in Weiner et al shows the effect of repetition on classification of category information. Third, Figure 9 shows the correlation between distributed response to repeated stimuli compared to distributed category-selectivity responses.

Likewise, except for one brief citation in the discussion, they still do not discuss Vogel's findings or conclusions in light of the current study at all. In reference to Vogels they write: "This insight enables us to reconcile fMRI repetition effects in the light of tuning width of neurons in Macaque IT being unaffected by repetition..."

This is a gross under-reporting of Vogels group's work that has already implicated local scaling as the mechanism for repetition suppression in face-selective areas. Discussion of the present study's findings should be placed in context and expanded to relate with these prior findings.

Reviewer #3 (Remarks to the Author):

The authors have completely addressed all the issues I raised in my original review, and the additional testing for generalization of the results strengthens the overall conclusions. The authors have also thoroughly addressed the issues raised by Reviewer 2, and I agree with their responses.

This is a very interesting manuscript that is sure to have impact on the field.

Just a couple of minor things to fix:

- 1) Page 3: "Indeed, a recent study of multi-voxel patterns by Kok et al. that prediction of upcoming stimuli..." – seems to be a word missing here.
- 2) Page 3: "In addition, Kok et al. did not specifically test improved stimulus classification could also be explained..." – also seems to be a word missing here.

Responses to comments from reviewer #2:

1) The experimental design is inadequate to examine the effects of repetition suppression because in each experiment they have only 2 conditions (Exp 1: faces/scrambled faces; Exp 2: 45 and 135 oriented grating). In order to make inferences as how repetition affects tuning curves (as illustrated in Figure 1) they need to include multiple stimuli in these experiments. Simply put, 2 stimuli are insufficient to determine tuning. One needs to have a range of stimuli for that as there are many possible curves that can occur between 2 points (reflecting their experimental conditions). Because they do not have data for other points, they cannot derive any inferences about tuning, selectivity, or the effect on repetition on these factors. Furthermore, several prior experiments in both monkey electrophysiology and human fMRI (e.g. from the Movshon, Kohn, Vogels, and Grill-Spector labs) have included multiple stimuli (e.g. multiple oriented gratings or motion directions or objects) to measure the effect of repetition on tuning curves. The only way to alleviate this key concern is to collect more data with more stimulus types for faces (e.g. using multiple levels of morphing from faces to scrambled faces) and more orientations for V1.

Because their data does not provide sufficient constraints to the model, their modeling work is insufficient for distinguishing between different neural accounts of repetition suppression

We understand the intuition of the reviewer that one needs to use multiple stimulus levels in order to evaluate how repetition affects tuning curves. Note firstly that one conceptual point of our paper is that the mapping from voxel tuning curves to neural tuning curves is not trivial (without a forward model), so one cannot directly “see” the effect of repetition on neural tuning curves with fMRI. One could of course use a forward model to simulate the effects of repetition on responses to multiple stimulus levels; however, as we argued in our previous response, the fact remains that our modelling clearly demonstrates that two levels for each stimulus type *are* sufficient to identify 1 out of 12 models as the only model able to explain the fMRI responses (and hence intuitions can be misleading). Nonetheless, we accept that more information might be important to distinguish more sophisticated models in future studies; a point we have now added to the paper:

page 8 – “Interestingly, our results show that fMRI experiments with just two levels for each stimulus type still enable one to differentiate between a wide range of possible models based on continuous neural tuning curves. However, it is possible that more than two stimulus levels will be needed to test simulations of repetition-related changes in neural tuning curves in future studies, e.g., to detect more subtle differences between different types of local scaling.”

2) Definition of face-selective activations. The authors contrast faces to scrambled faces and call the activations on the fusiform gyrus the FFA. This is problematic for two reasons: (i) many voxels in the fusiform gyrus and nearby regions on the occipital temporal sulcus respond more strongly to faces than scrambled faces, but that does not make them face-selective. For example, they may respond to bodies, words, or objects more than faces even as responses for faces are higher than scrambled faces. Specifically, next to the FFA there is both a body-selective region and a word-selective region. Why is this important for this experiment? This is important because their modeling hinges on that they know what is the tuning and the preferred stimulus for their voxels. But in fact, their measurements are not guaranteed to find the selective voxels.

It is indeed possible that some voxels in our “FFA” (and “OFA”) ROIs respond to other stimulus classes (such as bodies or words). However, we cannot see how this would change the conclusions of our simulations however. Furthermore, this might have occurred even if we had used objects, rather than scrambled faces, with which to compare faces (e.g, some voxels dominated by body-selective neurons might remain), or used both objects and bodies (e.g, some voxels dominated by word-selective voxels might remain), etc. In other words, one can never be certain that one’s non-face control stimuli are exhaustive enough to guarantee that only face-selective neurons remain.

(ii) The fact that they are calling their activations the FFA is misleading as they do not identify the FFA as Kanwisher does (faces vs. many other objects, see her 1997 and 2017 Journal of Neuroscience papers). Thus, it is unclear whether their activations are the same cortical region that Kanwisher calls the FFA. In other words, calling two different regions by the same name is misleading and adds confusion to the field.

We accept that our definition of FFA did not adhere to the one originally proposed by Kanwisher (nor subsequent definition of OFA adopted by others in the field). Therefore, we now refer to these areas as the fusiform face-responsive region (FFR) and occipital face-responsive region (OFR) throughout the manuscript, and have added a note that these do not conform to the precise definition used by Kanwisher and colleagues:

“The main a priori ROI was the face-responsive region in the fusiform gyrus (FFR), as defined by the group univariate contrast of unfamiliar faces versus scrambled faces (averaged across initial and repeated presentations, and therefore not biasing analysis of subsequent repetition effects), thresholded at $p < .05$ family-wise error corrected using random field theory. (While this region is likely to overlap with the Fusiform Face Area (FFA) defined by Kanwisher and colleagues³⁰, the FFA is normally defined for individual participants using a wider range of non-face control stimuli).”

3) Circularity of analysis. In the prior review, I have highlighted the dangers of circular analyses by using the same voxels to select the region and perform the main analysis of interest. The authors replied by writing that (1) I am wrong that they have a bias and (2) a 2006 paper co-authored by Henson, Friston, and colleagues argued why you should never use localisers. I disagree on both accounts. First, there is always a voxel-selection bias when using the same voxels to select a region and analyze its data. Second, in the same 2006 NeuroImage issue, Saxe, Kanwisher, and colleagues argued why it is always necessary to run functional localisers. Thus, at minimum they should acknowledge the other point of view. There are many ways in which they could have run a control to address the concern raised in the prior review. They could have used $\frac{1}{2}$ the data to select the voxels and $\frac{1}{2}$ the data to analyse repetition effects. Alternatively, they could have used an anatomical region to select voxels. This would have alleviated my concern. Arguing with a rhetoric that is more than a decade old and is not broadly accepted does not alleviate my concerns.

We were not trying to argue with rhetoric, and believe that the debate about functional localisers remains alive. Rather, we tried to explain intuitively (in terms of the sum and difference of two random variables) why there is no statistical bias when selecting voxels with an orthogonal (localizer) contrast (i.e., orthogonal contrast weights and orthogonal design matrix, as in the group-based localizer contrast used in our paper). To show this another way, we now enclose below some Matlab code that demonstrates that when one

selects voxels on the basis of a “face - scrambled” contrast (collapsed across repetition, i.e, using the contrast weights $c=[1\ 1\ -1\ -1]$ in code below), the probability of finding a significant effect of repetition in that subset of voxels (using orthogonal contrast weights for “initial - repeat”, either averaged across faces, $[1\ -1\ 1\ -1]$, or for faces $[1\ -1\ 0\ 0]$ or scrambled faces $[0\ 0\ 1\ -1]$ separately) does NOT change, regardless of whether a true repetition exists or not (i.e, does not exceed the nominal false positive rate of the statistical test, eg .05, when no true effect).

```

% Simulation to show that defining voxels based on one contrast does not
% bias univariate stats for a second orthogonal contrast
% (see generalised code for multivariate pattern test)

Nv = 1000; % initial number of voxels
Nt = 100;  % number of trials
SNR = 0.1; % signal-to-noise ratio

% Mean for 4 conditions (eg F1, F2, S1, S2 in Alink et al)
B = [2 2 1 1]'; % F > S, 1=2 (ie no repetition effect)
%B = [2 1 1 0]'; % F > S and 1>2 (repetition effect)

Nc = length(B);
Bpat = kron(kron(B,ones(Nt,1)),ones(1,Nv)); % base pattern across trials+voxels

X = kron(eye(Nc),ones(Nt,1)); % Design matrix for GLM below

bias = [];
for h=1:1000

    % Generate data and noise
    y = Bpat + randn(Nt*Nc,Nv)/SNR;

    % Evaluate significance of localising contrast at each voxel
    c = [1 1 -1 -1];
    [T,p] = fit_glm(X,y,c);

    % Select just voxels below some threshold (p-value)
    ind = find(p<.05); Nsv = length(ind); bias(h,1) = Nsv/Nv;
    y = y(:,ind);

    % y = mean(y,2); % average across voxels if want ROI stats (makes no difference)

    % Evaluate significance of test contrast on each voxel
    c = [1 -1 1 -1]; % orthogonal, so does not inflate false positive rate
    % c = [1 -1 0 0]; % orthogonal, so does not inflate false positive rate
    % c = [0 0 1 -1]; % orthogonal,so does not inflate false positive rate
    % c = [1 0 -1 0]; % correlated, so does inflate false positive rate

    [T,p] = fit_glm(X,y,c);

    ind = find(p<.05); bias(h,2) = length(ind)/Nsv;

    fprintf(' ');
end
fprintf('\n')

mean(bias)

figure,
subplot(1,2,1),hist(bias(:,1)),title('Localising Contrast'),
subplot(1,2,2),hist(bias(:,2)),title('Main Contrast')

function [T,p] = fit_glm(X,y,c)
% Code for any General Linear Model (eg T-test)

Bhat = pinv(X)*y;
Y = X*Bhat;
r = y - Y;
df = size(y,1) - rank(X);
s = r'*r / df; s = diag(s)';

T = c*Bhat ./ sqrt(s*(c*pinv(X'*X)*c'));
p = 2 * tcdf(-abs(T), df); % two-tailed

```

```
return
```

```
end
```

The above code demonstrates the lack of bias for univariate effects of repetition within each selected (localized) voxel. To confirm that there is no bias that affects any of our other (e.g. multivariate) repetition effects, we also simulated example datasets (trial by voxel matrices) that included patterns (extended code “test_localiser_multivariate.m” provided on OSF link with data: <https://osf.io/ph26y>) and ran them through exactly the same analysis code used to analyse the 6 repetition effects in the empirical data reported in the paper. When there is no true effect of repetition on the patterns, the repetition effects (for 1000 simulated participants) for voxels showing a significant univariate difference between faces and scrambled are shown below (cf Figure 3 of paper):

As can be seen, there is no evidence of any of the six repetition effects being significant. More importantly, when we calculate the distribution of repetition effects (slope estimates over 1000 random experiments on 20 participants) according to whether all voxels are included, or only the subset of voxels that show significant univariate repetition effect (e.g., from a localizer contrast), the results are shown below (blue bars with all voxels; yellow bars with just localized voxels):

Note that the central tendency of the all 6 repetition effects does not differ as a function of the voxels included (centred around zero in both cases). The distributions are wider for the localized voxels, simply because there are fewer of them, so the repetition effects are estimated less reliably. Thus there is a reduction in precision, but again, no bias from localizing with same data.

One subtlety in the above simulations is that not only must the contrasts ($c \cdot X$) be orthogonal, but the error term should be spherical (which is true if the error is

independent across conditions, as in above example of “randn”, but may not be true if the data come from the same subjects or trial estimates from the same run). In the software we used (SPM), any nonsphericity is estimated and the data and model prewhitened to ensure this is the case. However, this prewhitening does make further assumptions concerning similar error correlation across voxels (see http://www.mrc-cbu.cam.ac.uk/wp-content/uploads/2015/03/Henson_EN_15_ANOVA.pdf), so to be certain that there was no bias in our data, we also re-analysed our data:

We did try splitting each participant’s runs into two sets, and using one set to define FFR and the other set to test the 6 repetition effects. Unfortunately, the dramatic loss in power (either in defining FFR or testing repetition effects, depending on set size) meant that, although the pattern was similar, the effects did not always reach significance. Therefore we adopted an alternative “leave-one-subject-out” approach, in which we defined FFR based on 17 of the subjects, and tested repetition effects on the remaining 18th subject, then iterated over all 18 subjects. This ensures that the FFR definition and FFR testing are done on independent data. The results are shown below. By comparing with Figure 3 of the paper, one can see that the pattern of significant effects is unchanged.

4) Novelty. The authors should build on prior results rather than superficially cite them to artificially highlight the novelty of their study. It is necessary for developing knowledge in the field to accurately state what was done in prior papers and how their data relates to prior reports. For example, while they now cite some papers that were ignored in the prior version of the manuscript, these citations are cursory and do not engage with the findings or anchor their paper accurately compared to prior research.

The novelty in their study is the nice and comprehensive array of computational models. The novelty is not in that they are the first to examine both univariate and multivariate effects of repetition.

For example, the authors write: “A previous study by Weiner and colleagues (14) formally modelled the relationship between neuron-level and voxel-level repetition effects. This study was an important demonstration that neural scaling and neural sharpening can reproduce similar effects of repetition on the mean fMRI response across voxels (a “univariate” response), but did not consider the effect of repetition on patterns of fMRI responses across voxels (a “multivariate” response). Indeed, a recent study of multi-voxel patterns by Kok et al.(15) that prediction of upcoming stimuli (which may also arise from repetition) improved classification of those stimuli, and this was used to support the idea of neural sharpening.

However, neither study considered how the effects of repetition on a neural population depend on the difference between a stimulus property and the preference of that neural population for that property.”

This characterisation that none of the prior studies examined systematically the effect of repetition on distributed responses is inaccurate. First, Figures 4 and 6 in Weiner et al show the distribution of repetition effects across voxels. Second, Figure 8 in Weiner et al shows the effect of repetition on classification of category information. Third, Figure 9 shows the correlation between distributed response to repeated stimuli compared to distributed category-selectivity responses.

We accept that we did not accurately describe the prior study of Weiner and colleagues, which we have now corrected in the second paragraph of the introduction:

“A previous study by Weiner and colleagues¹⁴ formally modelled the relationship between neuron-level and voxel-level repetition effects. This study was an important demonstration that neural scaling and neural sharpening can reproduce similar effects of repetition on the mean fMRI response across voxels (a “univariate” response). They also assessed effects of repetition on fMRI activation patterns (“multivariate” responses). However, they did not determine whether multivariate repetition effects were best modelled by neural scaling or neural sharpening. This question is especially relevant given the recent claim by Kok et al.¹⁵ that the prediction of upcoming stimuli (which may also arise from repetition) causes neural sharpening based on the observation that prediction improves fMRI pattern-based stimulus classification. Here we overcome the limitation of these studies by considering a wider range of neural models and by evaluating a larger combination of both univariate and multivariate fMRI data features. This enables us to determine, for example, whether repetition effects on fMRI pattern information are specific to neural sharpening.”

Likewise, except for one brief citation in the discussion, they still do not discuss Vogel's findings or conclusions in light of the current study at all. In reference to Vogels they write: "This insight enables us to reconcile fMRI repetition effects in the light of tuning width of neurons in Macaque IT being unaffected by repetition..."

This is a gross under-reporting of Vogels group's work that has already implicated local scaling as the mechanism for repetition suppression in face-selective areas. Discussion of the present study's findings should be placed in context and expanded to relate with these prior findings.

We have added citation of a helpful review by Vogels on single-cell repetition effects in the Introduction and Discussion, and expanded on the points that Vogels makes in this review in our Discussion, as requested. Indeed, we have performed a new simulation that supports Vogel’s claim that stimulus-specific repetition effects do not simply reflect firing-rate dependent fatigue:

“As reviewed by Vogels¹⁸, firing-rate dependent response fatigue, e.g., a prolonged hyperpolarization that is intrinsic to the recorded neuron, is unlikely to explain the properties of the stimulus-specific repetition effects of the type described here. Indeed, when we simulated a simplified, two-parameter version of the local scaling model, in which the b parameter was fully determined by the initial firing-rate of neurons, we could no longer reproduce the present repetition effects (see Fatigue model section of Methods). It is possible that the repetition effects observed here in FFR, OFR and even

V1 are “inherited” from earlier stages in the visual processing pathway (i.e, arise in the inputs to these areas¹⁸), which potentially explains the need for a wider domain of adaptation (i.e, our additional b parameter).”

Responses to comments from reviewer #3:

The authors have completely addressed all the issues I raised in my original review, and the additional testing for generalization of the results strengthens the overall conclusions. The authors have also thoroughly addressed the issues raised by Reviewer 2, and I agree with their responses.

This is a very interesting manuscript that is sure to have impact on the field.

Just a couple of minor things to fix:

- 1) Page 3: “Indeed, a recent study of multi-voxel patterns by Kok et al. that prediction of upcoming stimuli....” – seems to be a word missing here.
- 2) Page 3: “In addition, Kok et al. did not specifically test improved stimulus classification could also be explained....” – also seems to be a word missing here.

This section has now been rewritten (see response to reviewer #2’s 4th comment) and no longer contain these mistakes.